# Co-option of a non-retroviral endogenous viral element in planthoppers

Hai-Jian Huang [1,2], Yi-Yuan Li[1,2], Zhuang-Xin Ye[1,2,3], Li-Li Li[1,2], Qing-Ling Hu[1,2], Yu-Juan He[1,2], Yu-Hua Qi[1,2], Yan Zhang[1,2], Ting Li[1,2], Gang Lu[1,2], Qian-Zhuo Mao[1,2], Ji-Chong Zhuo[1,2], Jia-Bao Lu[1,2], Zhong-Tian Xu[1,2], Zong-Tao Sun [1,2], Fei Yan[1,2], Jian-Ping Chen[1,2,3] ✉, Chuan-Xi Zhang [1,2] ✉ & Jun-Min Li [1,2] ✉

Non-retroviral endogenous viral elements (nrEVEs) are widely dispersed throughout the genomes of eukaryotes. Although nrEVEs are known to be involved in host antiviral immunity, it remains an open question whether they can be domesticated as functional proteins to serve cellular innovations in arthropods. In this study, we found that endogenous toti-like viral elements (ToEVEs) are ubiquitously integrated into the genomes of three planthopper species, with highly variable distributions and polymorphism levels in planthopper populations. Three ToEVEs display exon–intron structures and active transcription, suggesting that they might have been domesticated by planthoppers. CRISPR/Cas9 experiments revealed that one ToEVE in *Nilaparvata lugens*, NlToEVE14, has been co-opted by its host and plays essential roles in planthopper development and fecundity. Large-scale analysis of ToEVEs in arthropod genomes indicated that the number of arthropod nrEVEs is currently underestimated and that they may contribute to the functional diversity of arthropod genes.

Endogenous viral elements (EVEs) are sequences of viruses that have become integrated into the genomes of their host organisms and are then passed down vertically to subsequent generations. EVEs are widespread across various eukaryotes and serve as molecular fossils of past viral infections, playing an important role in the evolution of host genomes[1,2]. EVEs derived from retroviruses (ERVs) have been studied extensively since host genome integration is mandatory for their viral life cycle[3,4]. For instance, approximately 8% of the human genome is made up of ERVs, which reflects past infections with diverse retroviruses[5]. Surprisingly, an increasing number of studies have shown that the genomes of host organisms have also become endogenized with non-retroviral RNA viruses, including both single-stranded (positive and negative) and double-stranded RNA viruses, even though these viruses do not code for reverse transcriptase[6,7]. With the development of whole-genome sequencing, advanced bioinformatics approaches, and an increase in the discovery of novel exogenous viruses, non-retroviral endogenous viral elements (nrEVEs) have been successfully identified in the genomes of various eukaryotes, including animals (especially insects), plants, and fungi, over the past decade[7–9].

As external genetic material added to the host genome, EVEs are subjected to natural selection pressure from the host. If the integration is harmful to the host, the EVEs are likely to accumulate mutations and be eliminated during the process. Alternatively, the integrated viral sequences may be passed down vertically and functionally adopted (co-opted) by the host organism to serve additional beneficial functions[10,11]. EVEs scattered throughout eukaryotic genomes may provide resistance against exogenous viruses, as has been evidenced for ERVs and nrEVEs in various hosts[10,12]. Notably, it has been shown that ERV-encoded envelope proteins and Gag proteins can act as

[1]State Key Laboratory for Managing Biotic and Chemical Threats to the Quality and Safety of Agro-products, Institute of Plant Virology, Ningbo University, Ningbo 315211, China. [2]Key Laboratory of Biotechnology in Plant Protection of Ministry of Agriculture and Zhejiang Province, Institute of Plant Virology, Ningbo University, Ningbo 315211, China. [3]College of Forestry, Nanjing Forestry University, Nanjing 210037, China. ✉e-mail: jianpingchen@nbu.edu.cn; chxzhang@zju.edu.cn; lijunmin@nbu.edu.cn

restriction factors against exogenous retroviruses in many vertebrates, including humans, chickens, sheep, mice, and cats[13–16]. Among the antiviral roles of nrEVEs, the exogenous expression of endogenous bornavirus-like nucleoprotein-encoding elements (EBLNs) in ground squirrels efficiently suppressed polymerase activity and inhibited bornavirus replication[17]. The first antiviral role of insect nrEVEs was demonstrated in bees (*Apis mellifera*), where a host genome with an integrated 420-bp sequence derived from the Israeli acute paralysis virus (IAPV) results in host resistance to IAPV infection, although the molecular mechanisms are still unclear[18]. Another notable example is that the production of nrEVE-derived PIWI-interacting RNAs (piRNAs) can successfully control the replication of cognate viruses in mosquitoes[19–21]. Nevertheless, in most animals, the open reading frames (ORFs) of reported nrEVEs are disrupted, resulting in the generation of piRNAs. Only a limited number of nrEVEs with intact ORFs have been detected to be transcribed, leaving uncertainties about whether these transcripts are further translated into functional proteins[11,22].

Although the domestication of EVEs has provided numerous benefits for host biological functions, in addition to antiviral immunity, several co-opted EVEs have been repurposed to promote the development of novel cellular functions[10]. A prime example is the syncytin genes of vertebrates, which are the products of domesticated ERVs derived from multiple retroviral lineages and are essential for placental formation[23–25]. More recently, it has been shown that co-opted ERVs in mammalian genomes are involved in multiple functions in host innate immunity and mRNA delivery[26,27]. While there is increasing evidence to suggest that a number of ERVs have become integral components essential for host development and physiology, the co-opted novel cellular functions of nrEVEs in their hosts remain poorly understood, with the majority of work derived from the study of EBLNs[11,12]. Seven EBLNs (hsEBLN-1 to hsEBLN-7) have been identified in the human genome, with the transcript of hsEBLN-1 proposed to function as a long non-coding RNA that regulates the expression of an immune-related gene, COMMD3[28,29]. hsEBLN-2 was shown to be translated and encoded by a mitochondrial protein that is associated with cell viability, demonstrating the potential of co-opted mammalian function originating from ancient bornavirus infection[30]. Moreover, the EBLNs of miniopterid bats have been shown to encode an RNA-binding protein with biochemical properties similar to those of bornaviral nucleoprotein (N), suggesting that EBLNs can maintain the properties of their original genes[31]. In addition to EBLNs, a number of studies have shown that nrEVEs are also transcriptionally active, and several nrEVEs have maintained intact ORFs under strong purifying selection in invertebrates, mostly mosquitoes[7,32]. Although it has been suggested that nrEVEs are commonly found in piRNA clusters, these findings suggest that nrEVEs might undergo protein-level domestication and have biological functions[32–35]. However, there is currently no reliable evidence of cellular innovations serving host physiology and development derived from domesticated nrEVEs at the protein level, especially for invertebrates[10,11,19].

Viruses in the family *Totiviridae* consist of a single molecule double-stranded RNA (dsRNA) genome encoding a capsid protein (CP) and an RNA-dependent RNA polymerase (RdRp). The natural hosts of known totiviruses currently classified by the International Committee for the Taxonomy of Viruses (ICTV) are protozoa and fungi[36]. Endogenous toti-like viral elements (ToEVEs) were initially identified in three fungal genomes and were predicted to be maintained by purifying selection, with several ToEVEs able to produce transcripts[37]. However, candidate totiviruses were recently discovered in various invertebrates[38–43], leading to the successful identification of numerous ToEVEs in insects, crustaceans, nematodes, and others[9,44,45].

The brown planthopper (*Nilaparvata lugens* (Stål)), white-backed planthopper (*Sogatella furcifera* (Horváth)) and small brown planthopper (*Laodelphax striatellus* (Fallén)) are three of the most destructive insect pests in the rice field belonging to the insect family Delphacidae, order Hemiptera. Recently, the availability of chromosome-level genomes of the three planthoppers[46] and the discovery of novel totiviruses[41] provided an opportunity to investigate viral integration in these agriculturally important pests. In this study, ToEVEs in three planthoppers were identified and comprehensively analyzed. Importantly, we provide reliable and consolidated experimental evidence that one of the ToEVEs in *N. lugens* can be transcribed and translated into a functional protein related to insect development and fecundity, demonstrating the successful recruitment of a novel cellular function for arthropods from a tamed nrEVEs as the result of long-term host-virus co-evolution.

## Results

### Discovery of novel toti-like viruses in rice planthoppers

To identify potential toti-like viruses in planthoppers, we first performed a virome analysis by searching public Sequence Read Archive (SRA) datasets as well as newly generated transcriptomes from the three planthoppers using a collection of toti-like viruses as a query. As a result, three novel planthopper toti-like viruses with nearly complete genomes were identified, including one in *N. lugens* (SRA Accession: SRR19073262) and two in *S. furcifera* (SRA Accession: SRR11729951). The toti-like virus found in *N. lugens* has a genome length of 7037 nt and is named Nilaparvata lugens toti-like virus 1 (NlToLV1, accession: ON402804), while the other two viruses in *S. furcifera* have genome lengths of 5214 nt and 7573 nt and are named Sogatella furcifera toti-like virus 1 (SfToLV1, accession: ON402805) and Sogatella furcifera toti-like virus 2 (SfToLV2, accession: ON402806), respectively. All three toti-like viruses have the canonical *Totiviridae* organization, with two intact ORFs encoding a predicted CP and an RdRp, and reads of SfToLV2 were more abundant than those of the other two viruses (Supplementary Fig. 1a–c). A BLASTP homology search against the NCBI non-redundant database suggested that NlToLV1 and SfToLV2 are potential new members of the family *Totiviridae*. An RdRp-based maximum likelihood (ML) tree placed the three planthopper toti-like viruses in different clades of the family (Supplementary Fig. 1d). SfToLV1 clustered with members of the genus *Victorivirus* that naturally infect filamentous fungi[47], suggesting that SfToLV1 might be a mycovirus from a fungus infecting the planthopper (*S. furcifera*). SfToLV2 and NlToLV1 belong to different unclassified groups that mostly contain insect viruses, which might represent new genera in the family. It is noteworthy that SfToLV2 did not cluster with two previously reported totiviruses in *S. furcifera* (Sogatella furcifera totivirus 1 and 2, SfToV1 and 2)[41], despite being identified in the same host species (Supplementary Fig. 1d).

### Ubiquitous integration of ToEVEs in the genomes of three rice planthoppers

To systematically investigate ToEVEs in the genomes of three rice planthoppers, protein sequences of all publicly available exogenous toti/toti-like viruses were searched against (tBLASTn) the genomes of the three planthoppers locally. As a result, a total of 3, 9, and 22 ToEVEs were successfully identified in *S. furcifera* (SfToEVE1-3, length 696-2944 nt), *L. striatellus* (LsToEVE1-9, length 446-4019 nt), and *N. lugens* (NlToEVE1-22, length 504-10,814 nt), respectively (Supplemental Table 1). Planthopper ToEVEs are homologous to various regions of CP or RdRp, and the predicted ORFs are usually disrupted by frameshift mutations, possibly due to long-term co-evolution with the host insect. Almost all ToEVEs integrated into the three planthopper genomes shared the highest identities with the CP and RdRp of NlToLV1, and the only exceptions were NlToEVE13 and NlToEVE14 which were most closely related to the CP of SfToLV2 (Fig. 1a–c).

Most chromosomes (Chrs) of *N. lugens* (except Chr4, Chr8, and Chr9) contained at least one NlToEVE, with NlToEVE1, NlToEVE6, NlToEVE7, NlToEVE8, and NlToEVE18 corresponding to both CP and

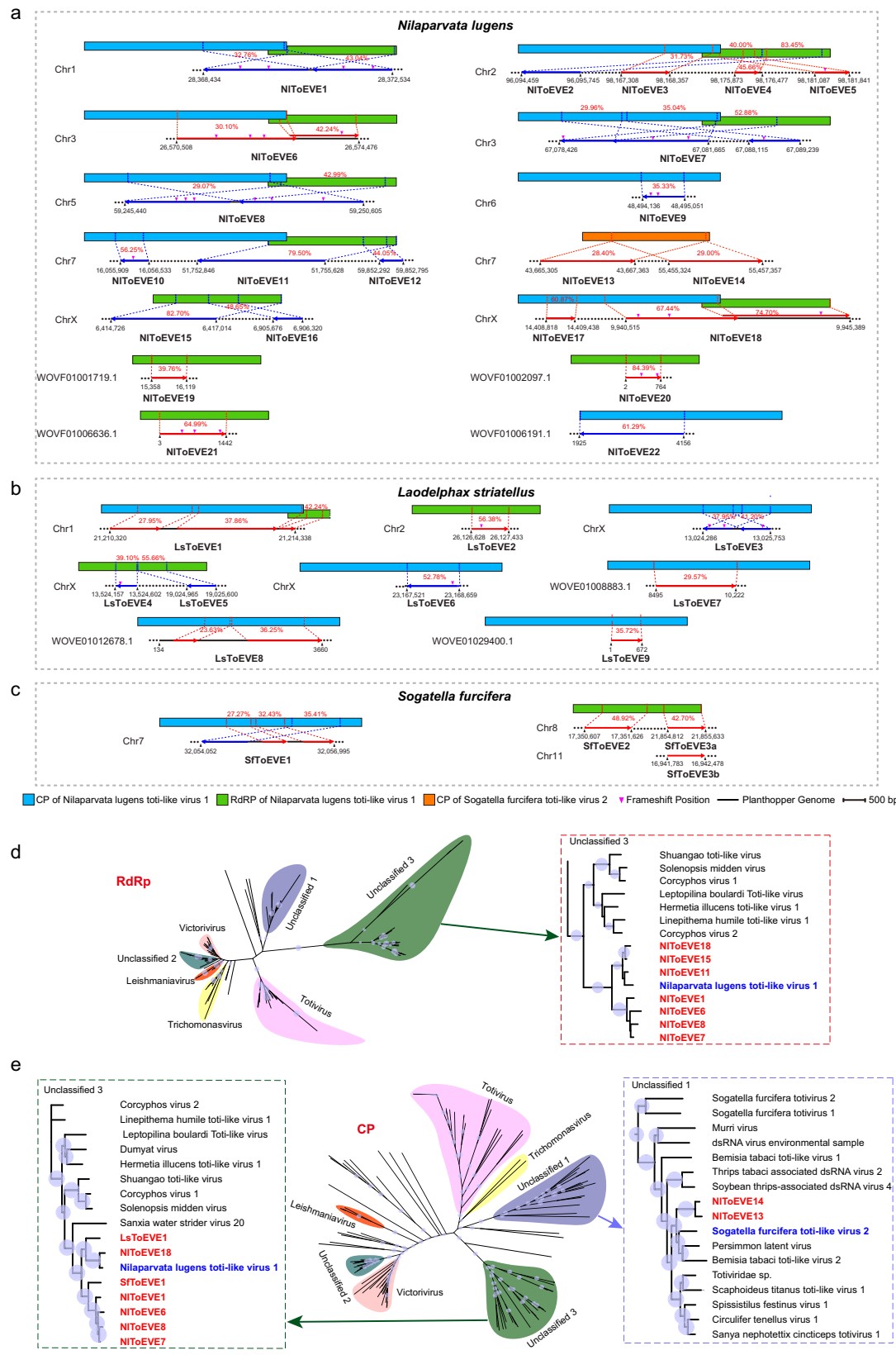

RdRp regions of NlToLV1, and others only shared similarity with one of the NlToLV1 genes. It is noteworthy that NlToEVE13 and NlToEVE14 have a predicted amino acid identity of 71.1%, and both of them correspond to the same CP region of SfToLV2, despite being far from each other on Chr7 (Fig. 1a). Similar scenarios were also observed for NlToEVEs on Chr2 (NlToEVE4-5) and ChrX (NlToEVE15-16), regardless

of the different regions of CP/RdRp to which they were mapped. Five ToEVEs in *L. striatellus* shared homology with the CP of NlToLV1, while LsToEVE1 corresponded to both CP and RdRp (Fig. 1b). In addition, four NlToEVEs and three LsToEVEs were identified in the unplaced scaffolds in the genomes of *N. lugens* and *L. striatellus*, respectively (Fig. 1a, b). Only three ToEVEs were identified in the genome of *S.*

**Fig. 1 | Endogenous toti-like viral elements (ToEVEs) in the genomes of three rice planthoppers.** The schematic diagram illustrates the identified ToEVEs within the genomes of *Nilaparvata lugens* (**a**), *Laodelphax striatellus* (**b**), and *Sogatella furcifera* (**c**). The cognate exogenous virus with the highest similarity is positioned above each ToEVE, and the corresponding homology regions (percentage of identities presented with red font) are shown with dotted lines. ToEVEs in the antisense strand are shown in blue lines, and red lines represent ToEVEs in the sense strand. Phylogenetic analysis of planthopper ToEVEs and other exogenous toti/toti- like viruses based on RNA-dependent RNA polymerase (RdRp) (**d**) and capsid protein (CP) (**e**) using the maximum likelihood algorithm. Nodes with bootstrap values > 50% are marked with solid blue circles, and the larger circles indicate higher bootstrap values. Taxonomic overviews of the viral family *Totiviridae* are shown on the left (RdRp tree) or in the middle (CP tree), and a close-up view of the clades of interest is shown in the dotted frames on the left or right side. Source data are provided as a Source Data file.

*furcifera* (Fig. 1c), although four exogenous toti/toti-like viruses have been discovered in this host insect (SfToV1, SfToV2, SfToLV1, and SfToLV2)[41]. Two SfToEVEs identified on Chr8 and Chr11 have >99% identity. They were named SfToEVE3a and SfToEVE3b and were considered the same SfToEVE, reflecting the possibility of recent totivirus integration and EVE replication in the insect genome. It is also note-worthy that NlToEVE3, NlToEVE4, and NlToEVE5 were located on the sense strand in Chr2 of *N. lugens*, while NlToEVE2 was located on the antisense strand of Chr2 (Fig. 1a).

To determine the taxonomic relationship of planthopper ToEVEs with contemporary toti/toti-like viruses, ML trees were constructed based on CP and RdRp protein sequences. As expected, in the tree based on the RdRp proteins, all of the selected NlToEVEs were clearly grouped with NlToLV1 and clustered together with other exogenous totiviruses in the clade "Unclassified 3" (LsToEVEs and SfToEVEs were not included in this analysis due to insufficient length of the predicted RdRp proteins) (Fig. 1d). For the tree based on the CP protein, most of the NlToEVEs, as well as LsToEVE1 and SfToEVE1, formed a well-supported monophyletic clade within "Unclassified 3" with NlToLV1 (Fig. 1e). Our orthologous analysis demonstrated that planthopper ToEVEs can be assigned to three orthologous groups, including group-1 (NlToEVE1, NlToEVE6, NlToEVE7, NlToEVE8, NlToEVE18, LsToEVE1, SfToEVE1), group-2 (NlToEVE11, NlToEVE15), and group-3 (NlToEVE13, NlToEVE14) (Supplemental Table 2). Moreover, a previous evolu-tionary study indicated that the three planthoppers (*N. lugens*, *L. striatellus*, and *S. furcifera*) used in this study belong to the same family Delphacidae. The species *S. furcifera* diverged from *L. striatellus* approximately 46.1 million years ago, whereas the common ancestor of these two planthopper species separated from *N. lugens* approxi-mately 64.4 million years ago[46]. Therefore, the wide ToEVEs integra-tion of orthologous group-1 in three planthopper genomes suggested that the ancient viral integration events might have occurred before 64.4 million years ago. In contrast, ToEVEs integration of orthologous group-3 was only detected in *N. lugens*, implying another independent viral integration event potentially occurred after 64.4 million years ago.

## Highly variable distribution and polymorphism level of ToEVEs in rice planthopper populations

Our previous study has investigated the migration of planthoppers based on the analysis of individual insect genome collected from dif-ferent countries of Asia and a site from north Australia[48]. To obtain an overview of the distribution of the identified ToEVEs among plan-thopper populations, ToEVEs were screened against these individual genomes of *N. lugens* (n = 256), *L. striatellus* (n = 28), *S. furcifera* (n = 18), and *N. muiri* (n = 2). The ToEVEs exhibited high variability (in terms of cover percentage) across the genomes of different individual plan-thoppers (Fig. 2a and Supplementary Fig. 2). For *N. lugens*, 256 indi-viduals were classified into 6 populations based on the inferred migratory trajectories[48]. NlToEVE9, 12-14, and 17-19 were present in almost all of the individual genomes with high cover percentages, whereas the distribution of other NlToEVEs was highly diverse. A similar distribution pattern was observed in various populations of *N. lugens* for each NlToEVE, except for the population of Australia (AUS), which exhibited a unique pattern (such as NlToEVE1, 9, 12, etc.)

(Fig. 2a). This difference is consistent with the findings of previous investigations on the migratory routes of *N. lugens* based on individual genomes. The Australian population was observed to exhibit sig-nificant genetic divergence and form a distinct group when compared to Asian populations. This difference can likely be attributed to geo-graphic barriers that may have limited gene flow and facilitated the divergence of the Australian population from the Asian populations[48]. Subsequent principal component analysis (PCA) confidently separated AUS from other geographic populations of *N. lugens* (Supplementary Fig. 3). It is worth noting that the three adjacent NlToEVEs (NlToEVE3- 5) on Chr2 (Fig. 1a) displayed a similar pattern in the individual gen-omes (Fig. 2a), and a piggyBac transposon (49.7% sequence similarity, e-value = 9e−77, sequence coverage= 99% to piggyBac transposable element-derived protein 3-like (XP_039275920.1)) was predicted between NlToEVE3 and NlToEVE4. In vertebrate, the herpesvirus was reported to be fused with a piggyBac-like DNA transposon and form a novel mobile element[49,50]. Therefore, it is possible that these NlToEVEs might have originated from similar viral insertion event in ancient times. *L. striatellus* and *S. furcifera* also showed a highly variable dis-tribution of ToEVEs in planthopper individuals (Fig. 2a), reflecting various evolutionary scenarios (positive selection or negative selec-tion) of these ToEVEs in insect genomes. *N. lugens* and *N. muiri* belong to the same genus in the family Delphacidae. Currently, high-quality genome of *N. muiri* is still not available and only genome resequencing reads from two individuals are available[48]. Therefore, all of the iden-tified planthopper ToEVEs were searched against the two individual genomes of *N. muiri*. The results showed that only NlToEVE13 and NlToEVE14 were detected with 98.7-100% coverage in *N. muiri* (Sup-plementary Fig. 2a). We proposed that the insertion event of these two NlToEVEs might have taken place after the divergence of *N. lugens* and the common ancestor of the other two planthopper species (*L. stria-tellus* and *S. furcifera*) after 64.4 million years ago and was stably inherited within the genome of the planthopper species in the genus *Nilaparvata*. To gain insight into the evolution of the identified ToEVEs in planthopper populations, the polymorphism level of ToEVEs for each genome was further estimated and compared with those of the fast-evolving genes (FEGs) and slow-evolving genes (SEGs) of the three planthoppers (*N. lugens*, *L. striatellus*, and *S. furcifera*). As a result, a highly variable polymorphism level was observed for the ToEVEs in the three planthopper populations (Fig. 2b), indicating that these ToEVEs might have evolved with the host genomes under various selective forces, as previously described[10,51,52]. Among the three species, NlToEVEs had a higher polymorphism level than the FEGs and SEGs, followed by LsToEVEs and SfToEVEs, which could be due to the dis-crepant number of individuals for the three planthopper species used in this analysis. In addition, the polymorphism levels of 21 ToEVEs (NlToEVEs3-5, 7-8, and 11-16; LsToEVEs2-7 and 9; and all of the SfToEVEs) were found to be comparable to those of the FEGs and SEGs in the three planthopper species (Fig. 2b), suggesting that these ToEVEs co-evolved with the host planthopper and may have con-tributed to host adaptation. Furthermore, the polymorphism level of NlToEVEs in five different geographic populations of *N. lugens* was found to be similar to the overall polymorphism patterns of NlToEVEs, except for the AUS population (Supplementary Fig. 2 and Fig. 2b), which is consistent with the PCA result (Supplementary Fig. 3).

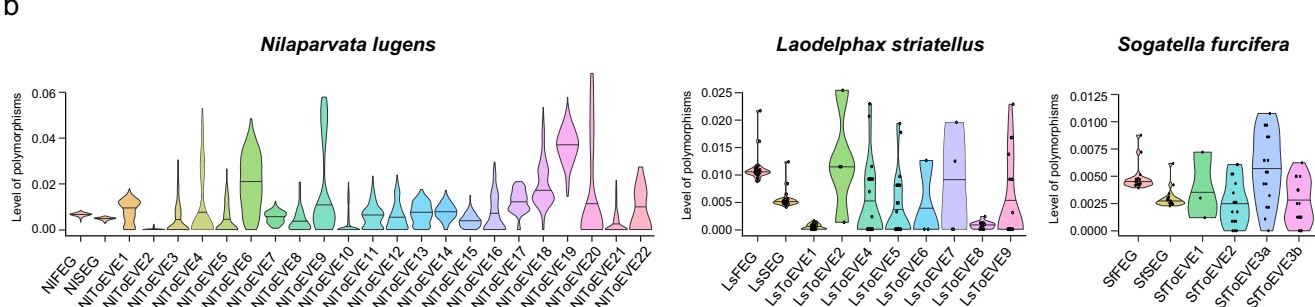

**Fig. 2 | Variable distribution and polymorphism level of ToEVEs in rice planthopper populations. a** ToEVEs exhibit high variability (cover percentage) across the genomes of different individual planthoppers of *Nilaparvata lugens* (NlToEVEs) (comprising six populations), *Laodelphax striatellus* (LsToEVEs) (1 population), and *Sogatella furcifera* (SfToEVEs) (1 population). Abbreviations of six *N. lugens* populations, AUS: Australia; BGD: Bangladesh; EA: East Asia; CN_FJ: Fujian, China; SA: South Asia; SEA: Southeast Asia; **b** Estimated polymorphism level for ToEVEs in the populations of *N. lugens*, *L. striatellus*, and *S. furcifera*. Abbreviations, NlFEG: Fast-Evolving Genes of *N. lugens*; NlSEG: Slow-Evolving Genes of *N. lugens*; LsFEG: Fast-Evolving Genes of *L. striatellus*; LsSEG: Slow-Evolving Genes of *L. striatellus*; SfFEG: Fast-Evolving Genes of *S. furcifera*; SfSEG: Slow-Evolving Genes of *S. furcifera*. Bars in violin plots correspond to the medians. *n* = 256, 28, and 18 individuals in *N. lugen*, *L. striatellus*, *S. furcifera*, respectively. Source data are provided as a Source Data file.

## Discrepant profiles of transcripts derived from ToEVEs in different populations and various developmental stages of planthoppers

To systematically investigate the potential transcription of the identified planthopper ToEVEs, we screened a total of 120 publicly available planthopper transcriptomes (43 *N. lugens*, 37 *L. striatellus*, and 40 *S. furcifera*) from various submitters (Supplemental Table 3) for ToEVEs derived transcript reads. Reads originating from 5 of the 22 NlToEVEs were detected across the 43 datasets of *N. lugens*, with the transcripts of NlToEVE13 and NlToEVE14 being ubiquitously and relatively highly expressed. In contrast, most LsToEVEs (6 out of 9)

were successfully transcribed in at least one dataset, such as LsToEVE6, which was expressed only in the datasets submitted by the Chinese Academy of Sciences (CAS) and Chinese Academy of Agricultural Sciences (CAAS) (Fig. 3a). For the three ToEVEs in *S. furcifera*, SfToEVE3 was ubiquitously expressed in all 40 screened datasets, and transcripts of SfToEVE1 were detected in most datasets, whereas SfToEVE2 was only present in the datasets provided by the University of Science and Technology of China (USTC) and China National Rice Research Institute (CNRRI) (Fig. 3a). The discrepant expression patterns of the same ToEVEs in various planthopper datasets suggest that these ToEVEs might be absent in some of the planthopper

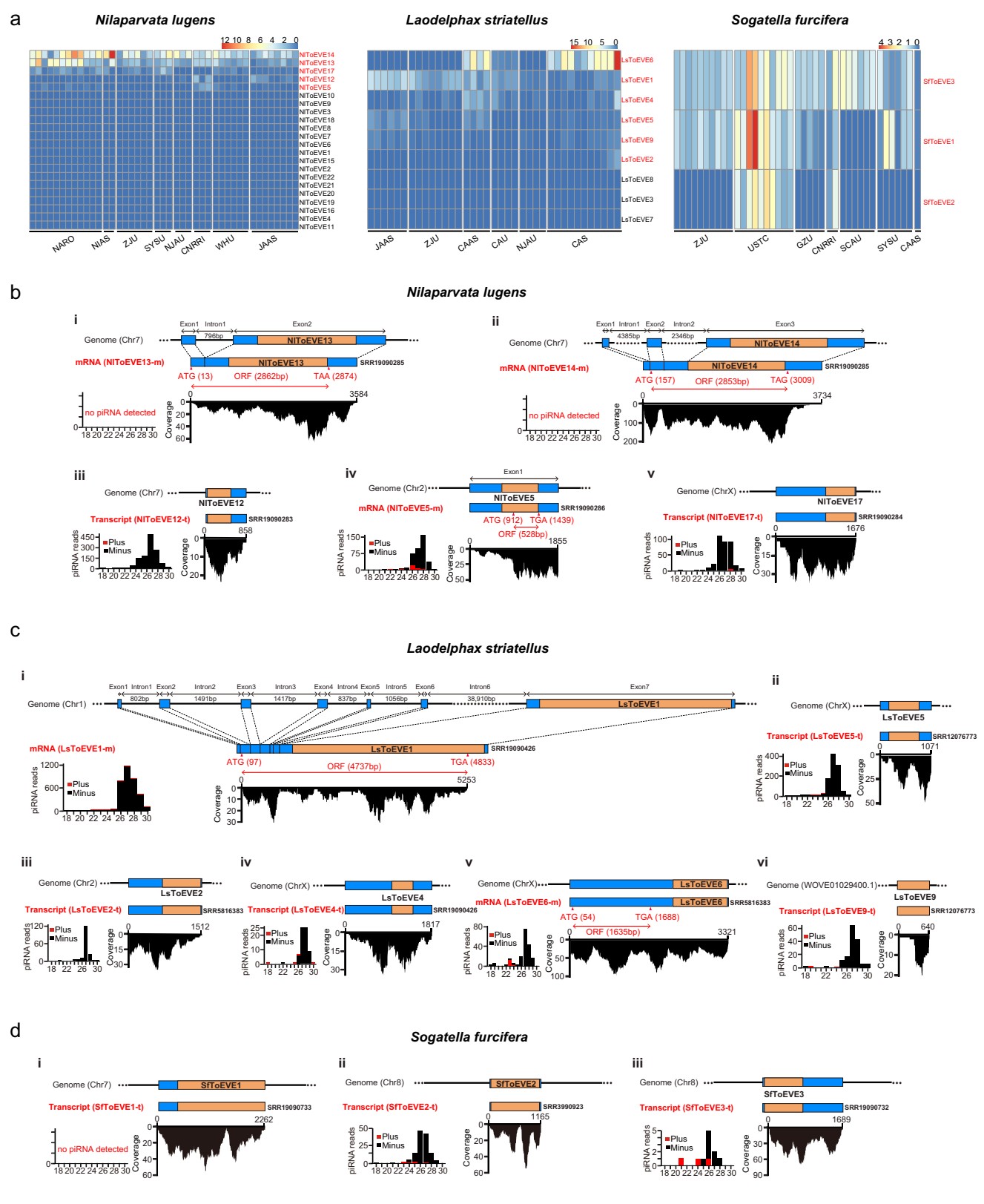

datasets, as illustrated in Fig. 2a. Alternatively, another explanation could be that the corresponding transcripts of these ToEVEs are only induced under specific circumstances. A number of transcribed ToEVEs in *N. lugens* (NlToEVE12-14, 17), *L. striatellus* (LsToEVE1, 4-5), and *S. furcifera* (SfToEVE2-3) were detected in all individual planthopper genomes with a high cover percentage (Fig. 2a), suggesting

that they might be stably integrated and inherited in planthopper genomes.

The expression profiles of ToEVEs in different developmental stages of the three planthopper species were investigated using our laboratory populations. As shown in Supplementary Fig. 4, disparate expression of ToEVEs was also observed in *L. striatellus* and *S. furcifera*,

**Fig. 3 | Transcription profiles of ToEVEs in different planthopper populations and schematic diagram representing transcripts containing ToEVEs in the genomes of the three planthoppers. a** Heatmap representing the abundance of transcript reads derived from ToEVEs of *Nilaparvata lugens*, *Laodelphax striatellus*, and *Sogatella furcifera* across planthopper datasets of various origins. Abbreviations of the datasets submitters: NARO, National Agriculture and Food Research Organization; NIAS, National Institute of Agrobiological Sciences; ZJU, Zhejiang University; SYSU, Sun Yat-sen University; NJAU, Nanjing Agricultural University; CNRRI, China National Rice Research Institute; WHU, Wuhan University; JAAS, Jiangsu Academy of Agricultural Sciences; CAAS, Chinese Academy of Agricultural

Sciences; CAU, China Agricultural University; CAS, Chinese Academy of Sciences; USTC, University of Science and Technology of China; GZU, Guizhou University; SCAU, Sichuan Agricultural University. Schematic diagrams represent the position and coverage of transcripts containing ToEVEs within the genome of *N. lugens* (**b**), *L. striatellus* (**c**), and *S. furcifera* (**d**). The size distribution of small RNAs derived from each of the ToEVEs is displayed on the left of the cover panel. The SRA accession numbers used for the analysis of the corresponding ToEVEs are provided to the right of the transcripts. Predicted open reading frames (ORFs) are indicated with red double-headed arrows. Source data are provided as a Source Data file.

with only two LsToEVEs (LsToEVE1 and LsToEVE4) and two SfToEVEs (SfToEVE1 and SfToEVE3) expressed ubiquitously in various developmental stages, while the expression patterns of NlToEVEs were similar to those in *N. lugens* populations from various origins (Fig. 3a). Relatively high numbers of reads derived from NlToEVE14 were detected compared to those of other NlToEVE transcripts, and it is intriguing to note that the expression of NlToEVE14 exhibits regular periodicity with a peak of approximately 24–48 h after molting in each developmental stage (Supplementary Fig. 4a). Moreover, comparatively elevated transcript levels of NlToEVE14, LsToEVE1, LsToEVE14, and SfToEVE1 were observed in the egg stage of *N. lugens*, *L. striatellus*, and *S. furcifera* (Supplementary Fig. 4), respectively, suggesting that these ToEVEs might be related to egg development in planthoppers.

### ToEVEs might be intrinsic parts of the planthopper intact mRNAs

To better understanding the transcription of ToEVEs, the potentially transcribed ToEVEs shown in Fig. 3a (red font) were chose. The corresponding transcriptomes with high abundant ToEVEs transcripts (Fig. 3a and Supplementary Fig. 4) were selected, assembled and further characterized (Fig. 3b–d). Most of the assembled planthopper transcripts were longer than the ToEVEs (except LsToEVE9, Fig. 3c-vi) and had relatively high coverage abundance (Fig. 3b-d). ORF prediction indicated that three NlToEVEs (NlToEVE5, NlToEVE13, and NlToEVE14) and two LsToEVEs (LsToEVE1 and LsToEVE6) were located within planthopper transcripts with intact ORFs ranging from 528 nt to 4737 nt (Fig. 3b, c). Furthermore, NlToEVE13, NlToEVE14 and LsToEVE1 were annotated within exons of planthopper genes and are expressed as part of these genes. The three ToEVEs (NlToEVE13, NlToEVE14, and LsToEVE1) were present in each of the detected individual genomes (Fig. 2a) and extensively expressed in all of the screened planthopper populations of various origins (Fig. 3a). Moreover, no additional domains were predicted in these ORFs other than the viral motifs. All of these observations provided convincing evidence that these ToEVEs might be co-opted by planthoppers and tamed as a group of novel genes with specific functions during long-term evolution.

In addition, piRNAs derived from nrEVEs have been commonly reported for a wide range of animals, including insects, which may serve as a reservoir of potential immune memory against cognate exogenous viruses[19,21,53,54]. In this study, small RNA (sRNA) sequencing was performed using the dissected ovaries of the three planthopper species. Analysis of ToEVE-derived sRNA profiles showed that the majority of the planthopper ToEVEs produced abundant sRNA reads with lengths ranging from 24 to 29 nt, which is typical of piRNAs (Fig. 3b–d), and these ToEVEs might serve as piRNA precursors. Interestingly, no piRNA was detected for two ToEVEs of *N. lugens* (NlToEVE13 and NlToEVE14) and one ToEVE of *S. furcifera* (SfToEVE1), despite the comparatively high transcript read numbers observed for these ToEVEs (Fig. 3b–i, ii, d–i).

### NlToEVE14 is translated as an authentic protein of *N. lugens*
Given the active transcription and predicted intact ORFs for the ToEVEs (Fig.3), it is intriguing to assess their potential translation in

planthoppers. We screened the public proteomic database and identified a single peptide (SPVYLLGDNSEIYMK) encoded by NlToEVE14 in one (PXD036431) of the three analyzed proteomic datasets (Fig. 4a). The specific antibody readily detected the protein band of NlToEVE14 at the expected size (predicted molecular weight of approximately 106 KD), and *NlToEVE14* dsRNA injection significantly reduced the expression of the target protein (Fig. 4b–c, indicated with red arrows), providing compelling evidence for the authentic translation of NlToEVE14 in *N. lugens*. Notably, no peptide derived from other planthopper ToEVEs was detected in this analysis.

We further investigated the tissue and developmental stage expression profiles of NlToEVE14 (both transcripts and proteins) in *N. lugens*. We observed an increase in the level of *NlToEVE14* transcripts during each developmental stage of *N. lugens* at 24-48 h after molting (Fig. 4d), consistent with the heatmap results (Supplementary Fig. 4a). The periodic expression of NlToEVE14 was also confirmed at the protein level for the 4th (peak at 36 h and 48 h after molting) and 5th (peak at 48 h and 72 h after molting) instars of *N. lugens* (Fig. 4e), indicating that NlToEVE14 may be associated with the development of *N. lugens*. Moreover, tissue expression profiles showed that NlToEVE14 was ubiquitously expressed in various planthopper tissues with relatively high abundance in the carcass (Car) and wing buds (WiB) at both the transcription and protein levels, as indicated in Fig. 4f, g, respectively.

### Essential roles of NlToEVE14 in *N. lugens* biology revealed by bioassays using CRISPR/Cas9 and RNA interference approaches
CRISPR/Cas9-mediated knockout of NlToEVE14 was conducted to investigate the potential roles of NlToEVE14 in the biological properties of *N. lugens*. Considering that NlToEVE14 is located in the C-terminal of the predicted ORF within the exon 3 (Fig. 3bii), the single guide RNA (sgRNA) was designed to target the boundary region between the planthopper-derived sequence and NlToEVE14 (Supplementary Fig. 5a). This design aimed to preserve the possible functionality of the *N. lugens* gene while eliminating the effects of viral integration. Two purified homozygous mutant strains (KO-M1 and KO-M2) were obtained for subsequent bioassays. The successful knockout of NlToEVE14 was validated by Western blotting, where the ~100 kDa band disappeared in both mutant strains, while the ~70 kDa nonspecific band (NB) could still be readily detected compared to the control (Supplementary Fig. 6). The population of KO-M1 (4 bp deletion, Fig. 5a) showed a significant extension in the duration of each nymph development stage and the total duration period (1st - 5th) of *N. lugens* (Fig. 5b), as well as reduced adult longevity for both females and males (Fig. 5c) compared to that in the wild type population (WT). No significant differences were detected in the survival rates of nymphs, the percentage of females, or the percentage of short-winged morphs between the KO-M1 and WT strains (Supplementary Fig. 7a–c). Fecundity analysis revealed a significant decrease in the number of eggs (Fig. 5d), the hatch rate (Fig. 5e), and the number of nymph offsprings (Fig. 5f), in KO-M1 compared to WT. Similar results were obtained for KO-M2 (8 bp insertion, Supplementary Fig. 8a) compared to WT (Supplementary Fig. 8b–i), providing strong evidence that

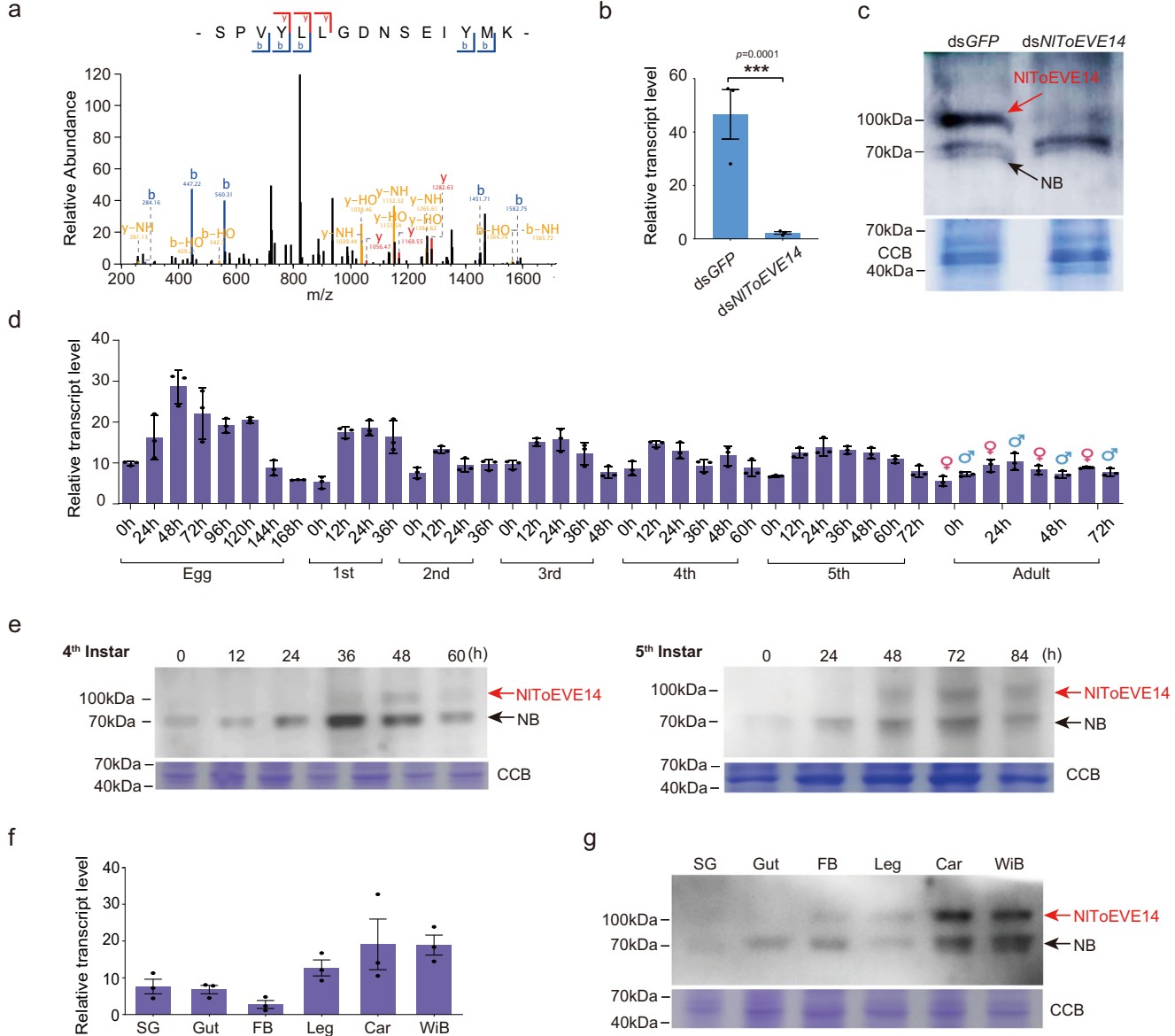

**Fig. 4 | Detection of NlToEVE14 protein and its expression profiles in different tissues and developmental stages of *Nilaparvata lugens*. a** Peptide of NlToEVE14 identified with proteomic analysis. **b** Efficient knockdown of *NlToEVE14* transcript with ds*NlToEVE14* injection (ds*GFP* was used as the control). *P*-values were determined by two-tailed unpaired Student's *t* test. ***P < 0.001. **c** The presence of NlToEVE14 protein was confirmed by Western blotting with a specific antibody. The band with the expected size of NlToEVE14 (~100 kDa) is indicated with a red arrow, whereas the non-specific band (NB, ~70 kDa) is marked with a black arrow. The ~100 kDa band became weaker after ds*NlToEVE14* treatment, while the ~70 kDa

band was not significantly affected. **d** Expression profiles of *NlToEVE14* transcripts in the eggs, nymphs (1st –5th instars), and adults (female and male) of *N. lugens*. **e** Protein expression profiles of NlToEVE14 in the 4th and 5th instar nymphs of *N. lugens*. Transcript (**f**) and protein (**g**) expression profiles of NlToEVE14 in different tissues of *N. lugens*. SG Salivary gland, FB Fat body, Car Carcass, WiB Wing buds. Data in **b**, **d**, and **f** are presented as mean values ± SEM (*n* = 3 independent biological replicates). The experiments in **c**, **e**, and **g** were repeated three times with similar results. Source data are provided as a Source Data file.

NlToEVE14 plays crucial roles in *N. lugens* biology. In addition to CRISPR/Cas9, dsRNA-mediated *NlToEVE14* knockdown was also conducted, which indicated a significant decrease in the number of eggs, whereas there was no difference in the hatch rate when planthoppers (3rd instar nymphs) were treated with ds*NlToEVE14* compared to the control (ds*GFP*) (Fig. 5g). The opposite effects were observed when newly emerged adult planthoppers were injected with ds*NlToEVE14* (Fig. 5h), suggesting that the duration of the knockdown effect (parental RNAi) is more distinct for adult treatment than for the nymph stage. Furthermore, ds*NlToEVE14* treatment of 1st instar nymphs significantly prolonged the duration of the nymph developmental stage without affecting the survival rate (Supplementary Fig. 9a, b), confirming CRISPR/Cas9 knockout results.

To gain a better understanding of the functions of NlToEVE14, Y2H screening was conducted using NlToEVE14 as bait to screen the cDNA library of *N. lugens*, resulting in the identification of a partial sequence annotated as glycine-rich cell wall structural protein 1 (NlTwd, XP_039289421). Further point-to-point Y2H assays showed that NlTwd interacted with NlToEVE14, specifically with the C-terminal of NlToEVE14 (NlToEVE14-C, 509-948aa, Supplementary Fig. 5b), which was subsequently confirmed by GST pull-down (Supplementary Fig. 10a, b). Similar to NlToEVE14, the expression profiles of NlTwd also showed a clear preference in the Leg, Car, and WiB tissues, with periodic expression patterns that had an accumulated transcript level in the later period of the egg and nymph stages (Supplementary Fig. 10c–d). Interestingly, the number of eggs and the hatch rate also

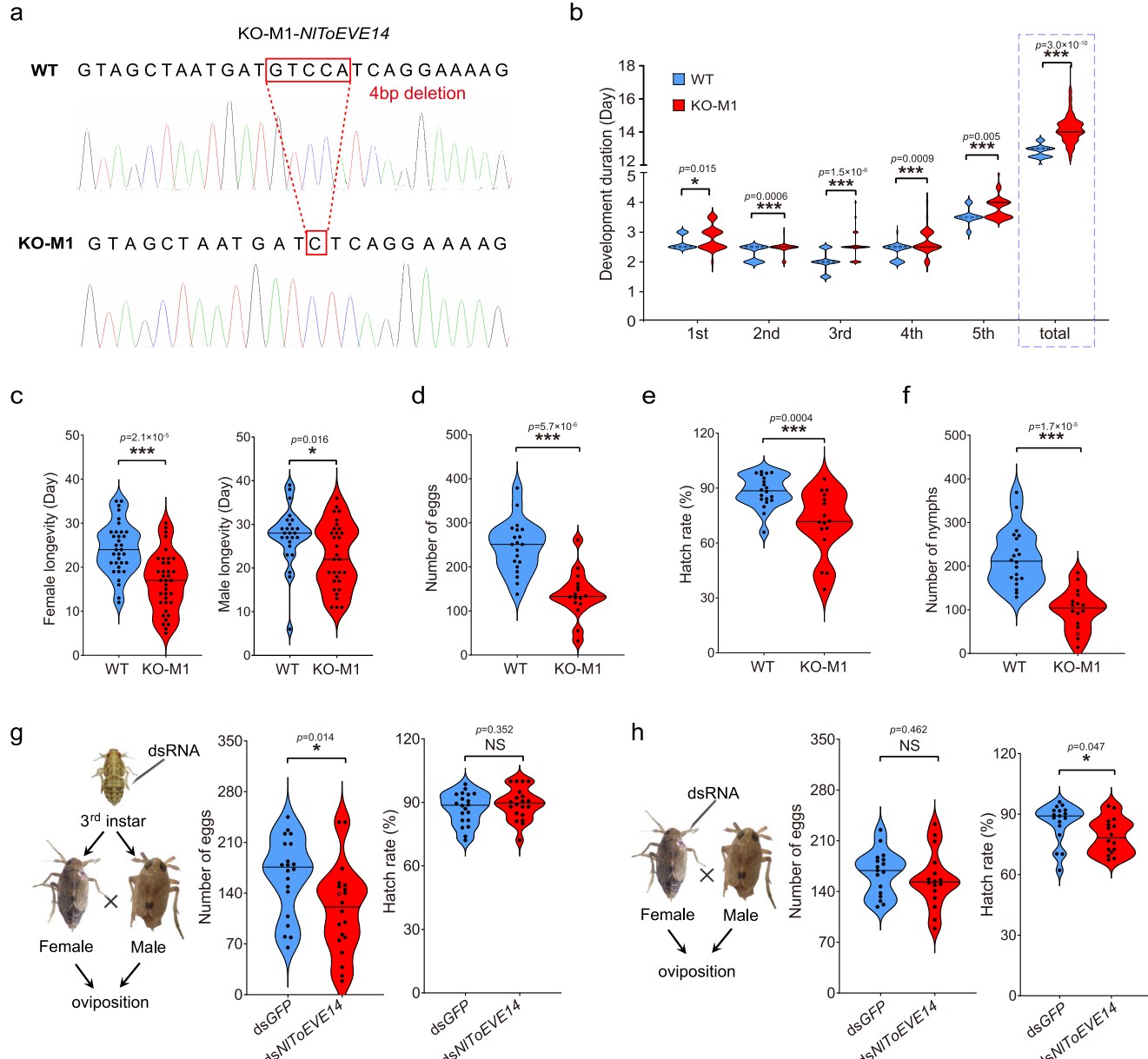

**Fig. 5 | NlToEVE14 is essential for the development and fecundity of *Nilaparvata lugens*, as revealed by CRISPR/Cas9 and RNA interference experiments. a** The KO-M1 strain, which is a homozygous mutant of *Nilaparvata lugens* for a 4 bp deletion in NlToEVE14. **b** The KO-M1 strain exhibited significantly prolonged nymph development stages compared to those of the wild-type strain (WT). *n* = 45 and 38 individuals in WT and KO-M1, respectively. **c** Knockout of NlToEVE14 (KO-M1) reduced the longevity of both male and female adult *N. lugens*. *n* = 34, 25, 37, and 31 individuals in WT female, WT male, KO-M1 female and KO-M1 male, respectively. In the KO-M1 strain, the fecundity of *N. lugens* was significantly decreased compared to that of the WT, including the number of eggs (**d**), the hatch

rate (**e**), and the number of nymph offsprings (**f**). *n* = 20 and 16 independent biological replicates in WT and KO-M1, respectively. The effects of NlToEVE14 knockdown on *N. lugens* fecundity when ds*NlToEVE14* was injected into individual 3rd instar nymphs (**g**) or adults (**h**) compared to the control (ds*GFP*). In **g**, *n* = 20 independent biological replicates in both ds*GFP* and ds*NlToEVE14*. In **h**, *n* = 19 and 16 independent biological replicates in ds*GFP* and ds*NlToEVE14*, respectively. Bars in violin plots correspond to the medians. *P* values were determined by two-tailed unpaired Student's *t* test. **P* < 0.05; ****P* < 0.001; NS not significant. Source data are provided as a Source Data file.

significantly decreased, while no difference was observed in the survival rate, upon treatment with ds*NlTwd* (Supplementary Fig. 10e–g). The interaction between NlToEVE14 and NlTwd, coupled with their similar expression profiles and knockdown effects, suggests that the functionality of NlToEVE14 might be associated with NlTwd in *N. lugens*.

Furthermore, the transcriptional patterns of the WT and KO-M1 strains were compared using transcriptomic sequencing. The results revealed 866 differentially expressed genes (DEGs), with 298 genes upregulated and 568 genes downregulated (provided in Source Data).

Gene Ontology (GO) analysis showed significant enrichment of DEGs related to the development and reproduction of planthoppers, including processes such as 'cuticle development', 'regulation of hormone levels', 'reproductive behavior', and 'structural constituent of chitin-based cuticle' (Supplementary Fig. 11a). Notably, the majority of cuticle-associated genes were found to be downregulated in the KO-M1 strain (Supplementary Fig. 11b). Additionally, the expression of the ecdysteroid biosynthesis gene CYP302A1[55] and a neuroendocrine convertase was significantly affected in the KO-M1 strain (Supplementary Fig. 11c), suggesting that hormonal pathways might also be

involved in the regulation of the planthopper phenotype in the mutant strain. However, the precise association of these DEGs with the various observed phenotypes after NlToEVE14 knockout remains unclear, and further investigation is needed to determine the exact functions of this tamed gene in *N. lugens*.

## Endogenization of ToEVEs in arthropod genomes is largely underestimated

The initial screening of 1188 publicly available arthropod genomes led to the identification of 5686 ToEVEs in 593 species of arthropods (Supplemental Data 1 and Supplemental Data 2). This finding indicates that ToEVEs are widely integrated into arthropod genomes. The majority of these ToEVEs were present in the class Insecta, with large numbers found in the orders Lepidoptera (136 species with 890 ToEVEs), Diptera (140 species with 990 ToEVEs), and Hymenoptera (176 species with 1827 ToEVEs), accounting for more than half of the identified ToEVEs (Supplementary Fig. 12). While the number of EVEs is known to be strongly correlated with the genome size and the genome quality of insects[22], the highly variable numbers of EVEs among different arthropod species remain unexplained. It has been observed that species with EVEs are often closely related to the host of EVE-homologous exogenous viruses, such as planthopper ToEVEs with planthopper-infecting totiviruses (present study) and mosquito flavivirus-derived EVEs with mosquito-infecting flaviviruses although most nrEVEs were derived from rhabdoviruses and chuviruses in *Aedes* mosquitoes[7,56]. To test this hypothesis, we selected 37 representative arthropod species (from Insecta, Arachnida, and Malacostraca) to explore the association between ToEVE-containing arthropods and the hosts of their corresponding totiviruses. The results showed numerous ToEVEs in the selected arthropods (Fig. 6a, Supplemental Table 4). As shown in Fig. 6b, counts of ToEVEs in specific species (left) were generally positively correlated with the corresponding families of these species harboring ToEVE-cognate totivirus (right), such as Delphacidae, Aleyrodinae, Figitidae, Culicidae, and Ixodidae. It is noteworthy that a large number of ToEVEs are homologous to the same exogenous totivirus (top hit), such as NlToLV1 (3 species with 32 ToEVEs), Leptopilina boulardi Toti-like virus (9 species with 36 ToEVEs), and Hubei toti-like virus 24 (2 species with 23 ToEVEs), whereas several totiviruses correspond to very few or no arthropod ToEVEs. The significantly different numbers of ToEVEs among the different cognate viruses can partially explain the variation in the number of EVEs identified in various arthropod species. This finding also suggests that the diversity of exogenous viruses is crucial for the discovery of novel EVEs and that the endogenization of ToEVEs in arthropod genomes may have been largely underestimated. Furthermore, the identified ToEVEs were searched against de novo assembled transcriptomes of the corresponding species to discover the potentially transcribed ones. Our results indicated that 57 ToEVEs representing 17 species were potentially transcribed, ranging from 202 nt to 11,718 nt (mean length 2153 nt) (Supplemental Table 5). This suggests that a number of ToEVEs might have been domesticated with potential functions in hosts similar to those of ToEVEs in planthopper genomes.

## Discussion

NrEVEs, a recently discovered type of EVE, have been characterized in various eukaryotic genomes, especially in insect genomes such as mosquitoes[6,11]. The abundance and distribution of nrEVEs are not homogeneous, as they depend significantly on the viral species and host genomes[7]. However, the potential functions and biological roles of evolutionarily co-opted nrEVEs, especially at the protein level, remain largely unknown at present[19,22]. In this study, we discovered totivirus-derived viral sequences, known as ToEVEs, in the genomes of rice planthoppers and further screened ToEVEs in the populations of three rice planthoppers. Our bioinformatic analysis and subsequent knock-out/knockdown experiments revealed that one ToEVE in *N. lugens*,

NlToEVE14, has been domesticated as a novel functional protein crucial for planthopper biology.

It is proposed that once a new nrEVE arises in the host genome, it will be exposed to evolutionary forces via the host species that are dependent on the fitness impact of the nrEVE on the host[10]. Only a small proportion of nrEVEs with beneficial, neutral, or slightly detrimental effects are retained in the host genome and may spread in the host population, whereas most deleterious nrEVEs are eliminated in a few generations[11,51]. Nevertheless, the prevalence of nrEVEs in host populations has been insufficiently investigated. In this study, a highly variable and heterozygous distribution of ToEVEs was detected in three planthopper populations (Fig. 2), similar to the distribution pattern of nrEVEs found in wild mosquitoes[32]. Our previous study, based on individual genomes of *N. lugens*, showed that the AUS population exhibited extensive genetic divergence from populations of Asian origin[48]. The presence/absence and PCA of NlToEVEs in *N. lugens* populations also clearly separated AUS from populations in other geographical regions (Fig. 2a and Supplementary Fig. 3), implying that EVEs and other host genetic materials might be under similar selection forces during evolution. Moreover, extensively diverse distributions of ToEVEs (presence/absence) in individual planthoppers were observed in the populations (Fig. 2a), such as NlToEVE2−5, 8, 10, 11, 15, 20-22, LsToEVE2, 3, 6-9, and SfToEVE1, suggesting that these ToEVEs might be under host selection forces. On the other hand, the high frequency of ToEVEs (NlToEVE9, 12-14, 17-19, LsToEVE1, 4, 5, and SfToEVE2, 3) might be the result of strong selection (Fig. 2a).

Transcriptionally active nrEVEs have been commonly found in arthropods, particularly in mosquitoes, as indicated by the detection of corresponding transcripts or nrEVE-derived piRNAs[34,35,53,57]. In addition to the transcription of DNA virus EVEs previously described in planthopper genomes[58], this study revealed that a number of ToEVEs were also diversely transcribed in different populations and various developmental stages of planthoppers (Fig. 3a and Supplementary Fig. 4). Previously, nrEVE sequences composed of complete or interrupted ORFs encoding various viral proteins were reported[12], and five intact ORFs were also predicted within transcripts of planthopper ToEVEs (Fig.3b, c). This is similar to the ORFs of nrEVE transcripts homologous to flaviviruses and bornaviruses previously determined in insect and mammalian species, respectively[6,59]. Intriguingly, exon–intron structures were discovered in the transcripts of planthoppers containing NlToEVE13, NlToEVE14, and LsToEVE1, which were located in the exon (Fig. 3b, c), resembling the typical feature of eukaryotic genes[60]. This phenomenon was also demonstrated for bornavirus-derived nrEVEs in afrotherians[61], indicating that these EVEs might potentially be tamed as authentic genes of the hosts. It has been proposed that intron gain and duplication are crucial steps in achieving functionality for horizontally transferred genes from bacteria to eukaryotes[62]. However, considering these three ToEVEs were exclusively located within the last exon of predicted planthopper ORFs (Fig. 3), raising the possibility that they might repurpose (modify, enhance, or diminish) the current functions of previously existing planthopper genes.

While the potential protein translation ability of nrEVEs has been extensively investigated in recent studies, it remains an open question whether nrEVE-derived mRNAs can be translated in arthropods[11,63]. This study used proteome analysis, together with RNAi and Western blot experiments, to convincingly prove that NlToEVE14 encodes a protein that is ubiquitously expressed in *N. lugens* (Fig. 4). In addition, the prevalence of NlToEVE14 in a large number of individual *N. lugens* (Fig. 2a), active transcription of NlToEVE14 transcripts in different populations (Fig. 3a) and various developmental stages of *N. lugens* (Supplementary Fig. 4), and no piRNA derived from NlToEVE14 transcripts (Fig. 3b-ii) offer consolidated evidence for the existence of the protein encoded by NlToEVE14. It is worth noting that an additional three ToEVEs (NlToEVE13, LsToEVE1, and LsToEVE6) derived from CP

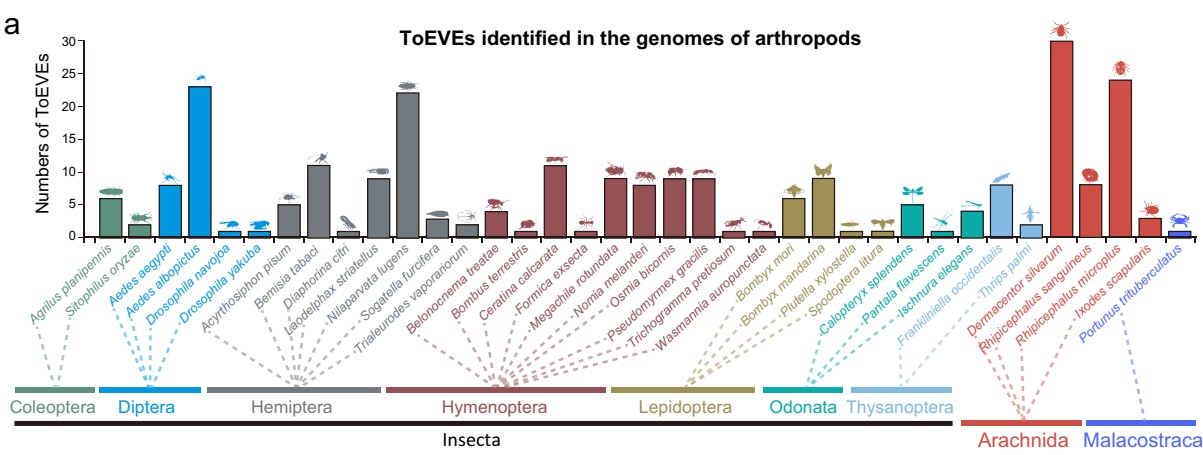

**Fig. 6 | ToEVEs identified in the genomes of representative arthropod species.**
**a** The number of ToEVEs in species from different orders in Insecta, Arachnida, and Malacostraca. **b** The association between ToEVE-containing arthropods and the host species of their corresponding cognate totivirus (top hit). Source data are provided as a Source Data file.

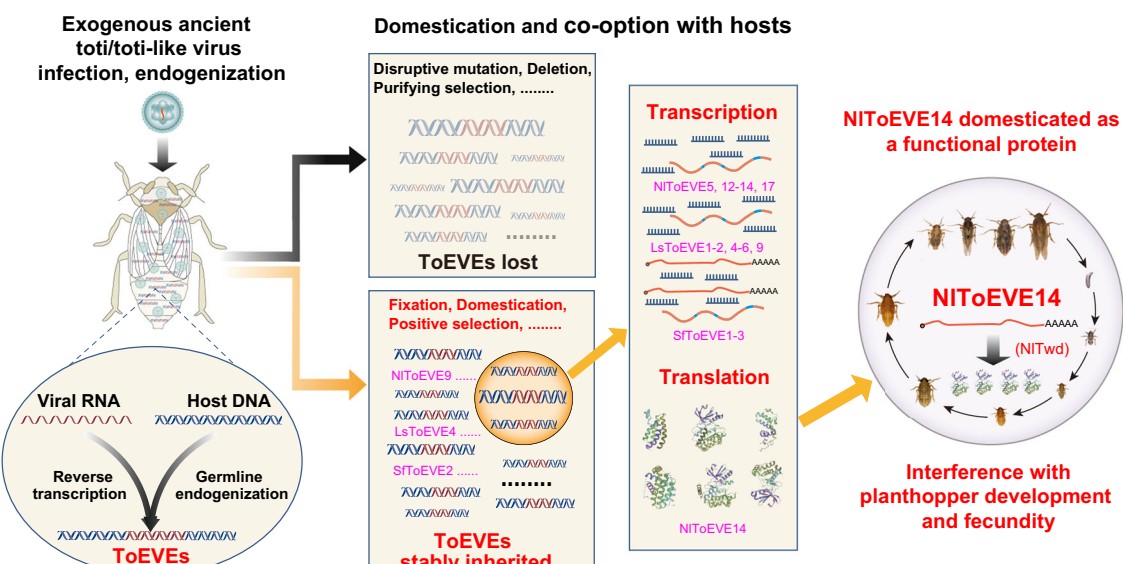

**Fig. 7 | A schematic diagram illustrating the long-term co-evolution between totiviruses and planthoppers and the novel cellular functions of *Nilaparvata lugens* that are derived from domesticated NlToEVE14. I** Replication of ancient exogenous toti/toti-like virus in planthoppers resulted in the endogenization of ToEVEs into the germ-line cells of the host chromosome. **II** The integrated ToEVEs underwent host selection forces during evolution, leading to them being either lost (detrimental effects) or stably inherited by the host genome and possibly spreading throughout the planthopper population (neutral or advantageous effects). **III** A number of the inherited ToEVEs are transcriptionally active with different forms of transcripts (small RNA, mRNA, etc.) or even translated into an authentic protein (NlToEVE14). **IV** The functional study provides convincing evidence that NlToEVE14 plays essential roles in the development and fecundity of *N. lugens*, suggesting that NlToEVE14 has been co-opted and serves as a novel host cellular protein.

and one derived from RdRP (NlToEVE5) of totivirus potentially contain ORFs. While only peptide of NlToEVE14 was identified in the screened proteomic datasets, it is important to note that the potential for translation of the other ToEVEs cannot be excluded, warranting further investigation.

Research on the biological roles of domesticated nrEVEs is still in its early stages, in contrast to the well-studied diverse functions of host co-opted ERVs. Evidence suggests that a portion of nrEVEs are transcriptionally active and produce piRNAs in arthropods, which could regulate cognate viral replication via sequence-dependent sRNA pathways[34,45,54,57]. However, in vivo experimental evidence for antiviral roles mediated by nrEVE-derived piRNAs against cognate viral replication was only revealed in *Aedes aegypti* ovaries[21]. In addition to the antiviral roles, functional studies of nrEVEs have mainly focused on co-opted EBLNs in mammals[28–31], whereas in vivo studies of the cellular functions of domesticated nrEVEs in arthropods are scarce. In this study, although the production of piRNAs was found for the majority of transcribed ToEVEs (Fig. 3b–d), whether these ToEVEs are associated with planthopper antiviral immunity against exogenous totiviruses requires further investigation. NlToEVE14 is a recombinant protein, with its N-terminus potentially derived from host insects and the C-terminal derived from viral integration. CRISPR/Cas9 knockout of the viral integration region resulted in prolonged development and decreased fecundity of *N. lugens* (Fig. 5). Additionally, NlToEVE14-C (509-948aa), which is located within the viral integration region (Supplementary Fig. 5b), is responsible for interacting with NlTwd, a cuticle-associated protein that is involved in insect development (Supplementary Fig. 10). These results suggest that an nrEVE (NlToEVE14) has been co-opted by the arthropod host and plays a role in insect physiology.

The application of the metatranscriptome approach has led to the discovery of enormous numbers of unexplored RNA viruses in arthropods[64–66], suggesting that arthropods serve as virome reservoirs and shape RNA virus evolution[67,68]. With the increasing number of newly identified toti/toti-like viruses, novel ToEVEs were recently found in arthropod species such as ticks[45], ants[44], and crustaceans[9,69]. Large-scale screening (1188 arthropod genomes) performed in this study led to the identification of 5686 ToEVEs scattered in the genomes of 593 arthropod species, mainly in the class Insecta (Supplementary Fig. 12). Analysis with selected representative arthropod species revealed that exogenous viruses are crucial for the discovery of novel EVEs (Fig. 6), and the ToEVEs potentially tamed in arthropods may be greatly underestimated at present. Additionally, conserved domains were barely identified within the predicted ORFs of ToEVEs in planthoppers and other arthropods, suggesting that they might be co-opted with hosts as a group of novel functional genes that may form an important component of the vast number of understudied proteins[70,71].

The long-term co-evolution between ToEVEs and planthoppers, as well as the cellular functions of *N. lugens* obtained from a domesticated NlToEVE14, is illustrated in a schematic diagram (Fig. 7). Endogenization of ToEVEs occurred occasionally when planthoppers were infected with an ancient toti/toti-like virus, and these ToEVEs may have been either lost or stably inherited by the host genome during co-evolution. A number of the inherited ToEVEs can be transcribed actively, and NlToEVE14 is further translated into a protein serving host cellular innovation that is crucial for planthopper biology (Fig. 7). Given the vast number of arthropod ToEVEs discovered in this study through large-scale screening, combined with the rapid identification of novel exogenous non-retroviral RNA viruses and improved high-quality arthropod genomes, it is likely that more domesticated nrEVEs with diverse novel cellular functions will be uncovered. While functional investigations of nrEVEs, especially at the protein level, are still in their early stage, these elements offer a broad range of new sequences within genomes that are subject to host selection pressure, enabling the emergence of novel repurposed functions. As seen in the case of ERVs, nrEVEs may also be critical to the functional diversity of genes in arthropods.

## Methods
### Insect cultures
The *N. lugens* and *S. furcifera* strains were originally collected in a rice field in Hangzhou (30.271˚N, 120.199˚E), China. The *L. striatellus* strain was originally collected in a rice field in Ningbo (29.90˚N, 121.63˚E)

China. The three rice planthopper strains were maintained separately in insect-proof cages on Nipponbare rice plants at 26°C ± 1°C, with a photoperiod of 16 h light : 8 h darkness and 70% ± 10% relative humidity in the laboratory of Ningbo University.

## Preliminary screen for the presence of nrEVEs with planthopper-infecting viruses

To investigate the potential integrations of planthopper viruses in the host genomes, a preliminary screen (tBLASTn) was conducted against the three planthopper genomes using a collection of planthopper-infecting viruses. This collection included Laodelphax striatellus iflavirus 1 (Accession: MG815140.1), Nilaparvata lugens honeydew virus-1 (Accession: NC_038302.1), Nilaparvata lugens honeydew virus-2 (Accession: NC_021566.1), Nilaparvata lugens honeydew virus-3 (Accession: NC_021567.1), Nilaparvata lugens reovirus (Accession: GCF_000852065.1), Sogatella furcifera honeydew virus 1 (Accession: MG818986.1), Sogatella furcifera totivirus (Accession: NC_040633.1), and Sogatella furcifera totivirus 2 (Accession: NC_040704.1). The results revealed that only totiviruses showed significant hits (e-value $< 1 \times 10^{-10}$) and were found to be integrated into the planthopper genomes. Consequently, totiviruses were chosen for the subsequent analysis of planthopper nrEVE in this study.

## Identification of novel planthopper toti-like viruses using meta-transcriptome

To investigate potential novel planthopper toti-like viruses, publicly available datasets (NCBI Sequence Read Archive (SRA) repository), as well as transcriptomes generated from our lab cultures, were analyzed. Potential toti-like viruses were discovered in N. lugens (generated in this study and deposited in SRA with accession SRR19073262) and S. furcifera (SRR11729951 retrieved from SRA). It is important to note that the species for the SRA dataset SRR11729951 should be S. furcifera rather than L. striatellus (annotated by submitter) which was confirmed by the analysis of the cytochrome oxidase subunit 1. For the transcriptome of N. lugens, a pool of approximately 20 planthoppers was used for total RNA extraction. After confirming the RNA integrity and quantity, the RNA samples were sent to Novogene (Beijing, China) for library construction and Illumina sequencing. Briefly, poly (A) + RNA was purified from 20 μg pooled total RNA using oligo(dT) magnetic beads. Fragmentation was conducted in the presence of divalent cations at 94 °C for 5 min; then, N6 random primers were used for reverse transcription to double-stranded complementary DNA (cDNA). After end-repair and adaptor ligation, the products were polymerase chain reaction (PCR)-amplified and purified using a QIAquick PCR purification kit (Qiagen, Hilden, Germany) to create a cDNA library[72]. The paired-end (150 bp) sequencing was performed on the Illumina HiSeq 4000 platform (Illumina, CA, USA). The raw reads of the RNA-seq sequences (datasets both from this study and SRA) were quality trimmed and assembled de novo using Trinity software v2.8.5 with default parameters[73]. All of the assembled contigs were compared with a customized local database comprised of all available toti/toti-like viruses retrieved from the NCBI protein database (retrieved on March 15th, 2022) using diamond BlastX locally[74]. Potential toti/toti-like viral contigs with high coverage (>20×), longer length ( > 2000 bp), and high homology to seed sequences (e-value $< 1 \times 10^{-20}$) were extracted and further confirmed by homology search against the NCBI nucleotide (NT) database. The sequences of the candidate viral contigs were then verified by reverse transcription PCR (RT-PCR) followed by Sanger sequencing. The primers used in this study are listed in Supplemental Table 6, and genome sequences of the three identified novel viruses were provided in Source Data and deposited in GenBank (ON402804 - ON402806). In addition, the discovered toti/toti-like viral contigs were annotated with InterPro 89.0[75], and a maximum likelihood (ML) tree based on RdRP protein sequences of totiviruses was constructed (1000 bootstrap replications) to evaluate the

taxonomical status of the viruses. Moreover, the coverage of the identified viruses was evaluated by realigning the RNA-seq reads back to the viral contigs. It should be mentioned that we noticed another toti-like virus deposited in GenBank, Fushun totivirus 2 (FuTV2, accession: MZ210014.1), which appears to be the same as one of the toti-like viruses identified in this work (accession: ON402805), except for the shorter genome length (5010 nt vs 5214 nt). Considering the shorter genome length and the fact that FuTV2 has not been characterized or published in any peer-reviewed journal, the totivirus identified in this study was also included for further analysis.

## Discovery and analysis of ToEVEs in the genomes of rice planthoppers

The chromosome-level genome assemblies of three planthopper species, N. lugens, L. striatellus, and S. furcifera, were retrieved from the NCBI genome database with the accession numbers GCA_014356525.1 (1088 M), GCA_014465815.1 (540 M), and GCA_014356515.1 (656 M), respectively[46]. Protein sequences of newly identified planthopper totiviruses, combined with the toti/toti-like viruses currently available in the NCBI protein database (retrieved on March 15th, 2022) were used as a query (provided in Source Data) and searched against the genomes of the three planthopper species using a tBLASTn algorithm with a cutoff E value $\leq 10^{-10}$. The potential ToEVEs were then extracted from the genomes accordingly and used to search against NCBI's entire protein database utilizing a reciprocal BLAST to eliminate false positive hits. The filtered planthopper ToEVEs were searched back to the customized database comprising the protein sequences of collected exogenous toti/toti-like viruses with the BlastX algorithm. Additionally, ML trees based on predicted proteins of RdRP and CP sequences derived from ToEVEs (containing complete or near complete RdRP/CP) and exogenous totiviruses (accession numbers provided in Source Data) were constructed using the method described above. The presence of the discovered ToEVEs in the genomes of the three planthopper species was confirmed by PCR followed by Sanger sequencing (primers listed in Supplemental Table 6). The identified ToEVE sequences of N. lugens, L. striatellus, and S. furcifera are provided in fasta format as provided in Source Data.

## Phylogenetic analysis of totiviruses and ToEVEs

All available toti/toti-like viruses, together with ToEVEs of the three planthopper species were used for construct the phylogenetic tree. The protein sequences of RdRP and CP were aligned with MAFFT v7.505[76], respectively, and ambiguously aligned regions were trimmed by Gblock v0.91b[77]. The best-fit model of amino acid substitution was evaluated by ModelTest-NG. Maximum likelihood (ML) trees were constructed using RAxML-NG with 1000 bootstrap replications[78]. Detailed information on the reference sequences used in the phylogenetic analysis is provided in the Source Data file.

## Orthologous analysis for protein datasets of the three planthopper species

Protein data from the three planthopper species were retrieved from InsectBase 2.0[79]. After filtering redundant alternative splicing events, the non-redundant protein data set was used to identify homologous pairs of sequences through the all-versus-all BLASTp algorithm with a significance cutoff of E-value $< 10^{-5}$. The BLASTp results were then converted into a normalized similarity matrix and processed using OrthoMCL v2.0.9[80] with default parameters (shortest protein length: 10; E-value $< 10^{-5}$). Protein families were identified using Markov chain clustering MCL14–137[81].

## Distribution of ToEVEs in the individual genomes of planthopper populations

To understand the distribution of these ToEVEs in natural populations of planthoppers, individuals of N. lugens, L. striatellus, S. furcifera, and

*N. muiri* derived from our preprint publication were used for whole genome resequencing and analysis[48]. The genome resequencing reads of *N. lugens*, *L. striatellus*, and *S. furcifera* were mapped to a collection of the identified planthopper ToEVEs using BWA MEM version 0.7.17[82]. For *N. muiri*, considering that there is no high-quality genome currently available for this species, ToEVEs identified in three planthoppers were searched against the genome resequencing reads of *N. muiri*. Considering that reference genomes might contain multiple identical ToEVEs, reads mapped to multiple positions were kept for downstream analysis. To evaluate the presence of ToEVEs, Mosdepth v0.3.3[83] was used to estimate per-base sequencing depth and sequencing depth using a 100 bp sliding window for visualization. Scripts for mapping and depth estimation are available on GitHub (https://github.com/lyy005/TotiEVEs, last accessed March 2, 2023).

The estimated average sequencing depth for the individual genomes of *L. striatellus* and *S. furcifera* was 31.5× and 35.0×, respectively. For ease of comparison between individuals, genomes with coverage less than 15.0× were excluded for the two planthoppers. Since the average sequencing depth for *N. lugens* was only 11.5×, the minimum sequencing depth for the individual genomes was set to 5.0×. Moreover, the presence of ToEVEs in the planthopper genome was considered only when the ToEVEs met specific sequencing depth and coverage criteria. For *S. furcifera* or *L. striatellus*, the ToEVEs were considered if they had a sequencing depth greater than 5 and coverage higher than 50%. As for *N. lugens*, the ToEVEs were included if they had a sequencing depth greater than 1 and coverage higher than 50%. In addition, for *N. muiri*, only two individual genomes were sequenced with a depth of approximately 65.0× (*N. lugens* was used as the reference genome). As a result, a total of 256, 28, 18, and 2 individual genomes of *N. lugens*, *L. striatellus*, *S. furcifera*, and *N. muiri*, respectively, were used to analyze the prevalence of ToEVEs in the individual genomes of different planthopper species (provided in Source Data). The 256 *N. lugens* individuals were classified into six populations based on their geographical origin, including Australia (AUS, $n = 6$), Bangladesh (BGD, $n = 25$), East Asia (EA, $n = 74$), Fujian China (CN_FJ, $n = 9$), South Asia (SA, $n = 72$), and Southeast Asia (SEA, $n = 70$)[48]. To examine the distribution of ToEVEs in different *N. lugens* populations, we performed principal components analysis (PCA) using R 3.5.

### Evolution of ToEVEs in the three planthopper populations

To gain a better understanding of the evolution of ToEVEs in planthopper populations, ToEVEs derived from individual planthopper genomes (containing at least 50% of the corresponding regions) were selected and the polymorphisms of ToEVEs were estimated using HaplotypeCaller in GATK version 4.2.6.1[84]. Then, a customized Perl script was used to estimate the polymorphism level of each ToEVE based on the VCF file of each resequencing individual (available at https://github.com/lyy005/TotiEVEs, last accessed March 2, 2023).

Additionally, to compare the evolutionary differences between ToEVEs and the planthopper intrinsic genes, the polymorphisms of fast-evolving genes (FEGs) and slow-evolving genes (SEGs) of the three planthoppers (*N. lugens*, *L. striatellus*, and *S. furcifera*) were also estimated following previous studies[32,52]. To identify FEGs and SEGs, briefly, the longest transcripts of all protein-coding genes for the three planthoppers were determined and assigned into orthologous groups using OrthoFinder version 2.5.4[85]. The 1,462 single-copy orthologs of the three planthopper species were then aligned with MAFFT LINSI version 7.505[76]. Poorly aligned genes (if more than 10% of the alignment regions are gaps) were removed using Gblocks version 0.91b[77]. The pairwise *p*-distance of each ortholog alignment was calculated, and the top 5% (73 genes) and the bottom 5% (73 genes) were selected as planthopper FEGs and SEGs, respectively (provided in Source Data). Moreover, polymorphism level of FEGs and SEGs were subsequently evaluated using the same method as described above for ToEVEs.

### Transcription profiles of ToEVEs in publicly available planthopper populations and different development stages of planthoppers

Public RNA-seq datasets of three rice planthoppers were retrieved and analyzed from the NCBI SRA repository to explore the potential transcripts of the planthopper ToEVEs. A total of 120 representative datasets (43 *N. lugens*, 37 *L. striatellus*, 40 *S. furcifera*) were selected based on the following criteria: data size over 3 Gb; removal of biological replicates (the dataset with the largest total number of bases was retained). Detailed information on these planthopper datasets is provided in Supplemental Table 3. The quality-trimmed raw reads of each dataset were mapped to the identified ToEVEs of the corresponding planthopper species with Bowtie2 v2.3.5.1[86] to investigate the transcript abundance of ToEVEs in planthopper populations of different origins.

To further investigate the developmental stage expression profiles of ToEVEs, ToEVE transcripts derived from raw reads were retrieved from transcriptomes representing different stages of the three planthopper species, which were generated from another project of our group. The transcriptomes were determined from egg, 1st instar, 2nd instar, 3rd instar, 4th instar, 5th instar, and male and female adults. Three, two, and three biological replicates were performed for each time point of *N. lugens*, *L. striatellus*, and *S. furcifera*, respectively. The relative abundance of ToEVEs in each sample was normalized as fragments per kilobase of transcript per million mapped reads (FPKM) and average counts were used to quantify and compare the expression at each time point. Information on RNA-seq reads corresponding to the ToEVE transcripts of the three planthoppers is provided in Source Data.

### Analysis of potential ToEVE transcripts in genomes of the three planthoppers

To identify transcripts containing ToEVEs in planthoppers, raw reads from publicly available datasets and transcriptomes generated in our lab were de novo assembled/reassembled using Trinity software[73]. The assembled contigs were then searched against a local customized database, which comprised all identified planthopper ToEVEs, using BlastN to obtain the ToEVE transcripts as provided in Source Data. To confirm the location of ToEVE transcripts within planthopper genomes accurately, the sequences of the identified ToEVE transcripts were extracted from the planthopper transcriptomes and used as a query to search against the corresponding planthopper species' genome. The matched region of ToEVEs in planthopper genomes was retrieved and extended by 2000 (or more) bases at both the 5′ and 3′ termini to predict open reading frames (ORFs) with the online ORF Finder server (https://www.ncbi.nlm.nih.gov/orffinder). The abundance of ToEVEs was measured by realigning the quality-controlled transcriptome raw reads back to the planthopper ToEVE transcripts. Finally, the ToEVE transcripts in planthoppers (adults, confirmed for the absence of totivirus infection) were verified by RT-PCR, followed by Sanger sequencing (primers listed in Supplemental Table 6).

### sRNA profiles of planthopper ToEVE transcripts

To investigate the possible presence of sRNAs derived from ToEVEs, ovaries were dissected from female insects. A pool of approximately 20 ovaries was used for total RNA extraction. cDNA libraries for each of the three planthopper species were constructed using the Illumina TruSeq Small RNA Sample Preparation Kit (Illumina, CA, USA), and sRNAs were subsequently sequenced on an Illumina HiSeq 2500 by Novogene (Tianjin, China). The raw reads of sRNAs were quality-controlled to remove the adapter, low-quality, and junk sequences, and clean sRNA reads with a length of 18-30 nt were extracted with FASTX-Toolkit v0.0.14 (http://hannonlab.cshl.edu/fastx_toolkit). The sRNA reads were then mapped to planthopper ToEVE transcripts using

Bowtie software v1.2.3 with a perfect match[87]. The subsequent analyses were performed using Linux bash scripts.

## Proteomic analysis for potentially translated ToEVEs in planthoppers

To determine if ToEVEs in planthoppers can be translated, LC-MS/MS-based proteomic data were retrieved and subsequently analyzed from a public proteomic database, including a *L. striatellus* dataset (ProteomeXchange accession: PXD023965) and three *N. lugens* datasets (ProteomeXchange accession: PXD036431, PXD043983, and PXD044065). For the *L. striatellus* dataset, the Mascot search engine (Matrix Science, London, UK; version 2.3.02) was utilized for searching potential ToEVE-encoded peptides, with the following parameters set: iTRAQ8plex for quantification, one missed cleavage tolerance of trypsin, monoisotopic mass accuracy, carbamidomethyl (C), iTRAQ8-plex (N-term), and iTRAQ8plex (K) as fixed modification, oxidation (M), and iTRAQ8plex (Y) as variable modification. In MS/MS mode, the fragment ion mass accuracy was set to <0.1 Da. In MS/peptide mode, the peptide mass accuracy was set to <0.05 Da. In addition, for the *N. lugens* datasets, the proteomic data was searched for potential ToEVE-encoded peptides using MaxQuant v1.6.5.0 with default parameters, including one missed cleavage tolerance of trypsin, carbamidomethyl (C), oxidation (M), and Acetyl (Protein N-term). Identifications were filtered to a 1% false discovery rate (FDR) at the peptide-spectrum match (PSM) level.

## Transcript profiles of *NlToEVE14* in *N. lugens*

To investigate the developmental expression profiles of *NlToEVE14*, raw reads from *NlToEVE14* transcripts were obtained from the transcriptome of *N. lugens*, as described above. The relative abundance of *NlToEVE14* was then evaluated in different developmental stages of *N. lugens*. For the tissue expression profiles of *NlToEVE14*, various tissues (including salivary gland, gut, fat body, leg, carcass, and wing buds) of *N. lugens* were collected. Total RNA from the collected samples was extracted using TRIzol Reagent (#10296018, Invitrogen, Carlsbad, CA, USA). according to the manufacturer's instructions. The transcripts of *NlToEVE14* were then determined by quantitative real-time PCR (qPCR) using the SYBR Green Supermix Kit (#11202ES08, Yeasen, Shanghai, China) and a Roche Light Cycler® 480 Real-Time PCR System (Roche Diagnostics, Mannheim, Germany). The PCR procedure was as follows, denaturation for 5 min at 95 °C, followed by 40 cycles at 95 °C for 10 s and 60 °C for 30 s. The primers used in qPCR were designed using Primer Premier v6.0 (Supplemental Table 6). Three independent replicates were performed for this experiment, and each replicates contained the tissues derived from approximately 40–50 individual 5th instars of *N. lugens* exactly 48 h after molting.

## Protein detection of NlToEVE14 in *N. lugens*

The protein level of NlToEVE14 was determined in *N. lugens* of different developmental stages and tissues. To collect various developmental samples, 3rd and 4th instar nymphs were reared on rice seedlings and used to obtain the newly emerged 4th and 5th instar nymphs, respectively. The newly emerged nymphs were then further maintained in a climate chamber and collected at every 12 h (4th instar nymph) and 24 h (5th instar nymph) intervals, respectively. Tissue samples of planthoppers were collected as described above. All samples were homogenized in RIPA Lysis Buffer (#89900, Thermo Fisher Scientific, Waltham, MA), and protein concentrations were quantified using a BCA Protein Assay Kit (#CW0014S, CwBiotech, Taizhou, China) following the manufacturer's instructions. After adding 6 × SDS loading buffer, the samples were boiled for 10 minutes. Proteins were separated by SDS-PAGE and transferred to PVDF membranes. The anti-NlToEVE14 serum, prepared by immunizing rabbits with purified His-NlToEVE14 (260-529aa) proteins, was produced via the custom service of HuaAn Biotechnology Company (Hangzhou, China). The anti-NlToEVE14 serum was diluted at 1:5,000, followed by additional incubation with horseradish peroxidase-conjugated goat anti-rabbit IgG antibody (1:10,000, #31460, Thermo Fisher Scientific, Waltham, MA). Images were acquired by an AI 680 image analyzer (Amersham Pharmacia Biotech, Buckinghamshire, UK). To monitor equal protein loading, samples were further stained with Coomassie brilliant blue. The full scan results of blots and gels were provided in Supplementary Fig. 13 and Source Data file.

## CRISPR/Cas9-mediated knockout of NlToEVE14

The potential target sites for synthesizing sgRNA of *NlToEVE14* were predicted using the sgRNAcas9 algorithm 3.0.5. The searching parameters were set as 20-nt in sgRNA length, 20-80% in GC content, and NGG for the PAM. Using these criteria, one candidate target sequence with the lowest off-target possibility (5′-GGTAGCTAATGATGTCCAT-CAGG-3′) was selected. PCR was performed using a forward primer containing the T7 sequence and a reverse primer containing the partial sgRNA sequence (Supplemental Table 6). The sgRNA was prepared using a T7 High Yield RNA Transcription Kit (#TR101-01, Vazyme, Nanjing, China) according to the manufacturer's instructions. The Cas9 mRNA was prepared using the mMESSAGE mMACHINE SP6 Transcription Kit (#AM1340, Thermo Fisher Scientific, Waltham, MA) and Poly(A) Tailing Kit (#AM1350, Thermo Fisher Scientific, Waltham, MA). Microinjection was performed following the method described previously[88]. In brief, a mixture of sgRNA (300 ng/ml) and Cas9 mRNA was injected into newly deposited eggs using the FemtoJet microinjection system (Eppendorf, Hamburg, Germany). The injected eggs were then transferred carefully to filter papers, which were rinsed with sterilized water containing tebuconazole (20 ng/ml) and kanamycin (50 ng/ml), and placed in a dark incubator at 26 ± 0.5 °C with a humidity level of 50 ± 5%.

## Purification of NlToEVE14 homozygous mutant populations for *N. lugens*

Approximately 10 days after injection, the hatched nymphs were carefully transferred to fresh rice seedlings and reared to the adult stage. The wings of newly emerged adults were carefully detached using forceps under a stereoscope, and genomic DNA was extracted from the wings using the Wizard Genomic DNA Purification Kit (#A1120, Promega, Madison, WA, USA). Potential mutations in individual planthoppers were examined by PCR followed by Sanger sequencing.

The G0 mutant individuals were paired with wild-type *N. lugens* to examine mutations in their G1 offspring, which were further paired to collect the homozygous G2 mutant stains for subsequent bioassays.

## Insect bioassays for the NlToEVE14 mutant strains of *N. lugens*

To investigate the potential biological functions of NlToEVE14, insect bioassays were performed to compare differences between NlToEVE14 mutants and wild strains of *N. lugens*. For survival and developmental duration analysis, newly hatched 1st instar nymphs were individually reared on 4–5-leaf stage rice seedlings, and the survival rates and stage durations were recorded every 12 h. For adult longevity analysis, newly emerged male or female adults were individually reared on 4–5-leaf stage rice seedlings, and the death time was recorded every day. Meanwhile, the female-male ratio and proportion of short-winged/long-winged morph was also recorded simultaneously. For fecundity analysis, the newly emerged adults were paired and allowed to oviposit for 10 days. The number of hatched offspring and dead embryos was counted. The sterile females were excluded to calculate the mean. For each strain, 40-60 individual insects were used for survival, developmental duration, and adult longevity analysis; 150-200 individual insects were used for female-male ratio analysis; 200-300 individual insects were used for short-winged/long-winged analysis; 15-20 individual insects were used for fecundity analysis.

### RNAi-mediated gene silence (knockdown) of NlToEVE14

The DNA sequence of *NlToEVE14* was amplified using primers (Supplemental Table 6) ligated with a T7-promoter sequence, and cloned into pClone007 Vector (#TSV-007, Tsingke, Beijing, China), with green fluorescent protein (GFP) as the control. The PCR-generated DNA templates containing T7 sequences were used to synthesize the dsRNAs with a T7 High Yield RNA Transcription Kit (Vazyme). The RNA interference (RNAi) experiment was conducted as previously described[89]. Briefly, the newly emerged 3[rd] instar nymphs or adults (for fecundity experiments) and newly emerged 1[st] instar nymphs (for survival and developmental duration experiments) were anesthetized with carbon dioxide for 5–10 s. Then, ds*NlToEVE14* was injected into the mesothorax of individual *N. lugens* using a FemtoJet (Eppendorf). Afterward, the injected insects were kept on the 4-5-leaf stage rice seedlings for 24 h and the living insects were selected for further investigation. Insect bioassays of survival, developmental duration, and fecundity were performed as described above.

### Two yeast hybridization

The Y2H screening assay was performed as follows: the complete coding sequence of *NlToEVE14* was cloned into the pGBKT7 vector, which was then used as bait to screen a normalized *N. lugens* cDNA prey library according to the manufacturer's instructions. Positive clones were selected on quadruple dropout (QDO) medium (SD/−adenine/−histidine/−leucine/−tryptophan), and prey plasmids were isolated from positive clones for Sanger sequencing. The Y2H point-to-point assay was used to investigate the interactions between NlTwd and different deletion mutants of NlToEVE14. Briefly, NlTwd and NlToEVE14 mutants were cloned into pGADT7 or pGBKT7 vectors, respectively. The recombinant vectors, along with the corresponding empty vectors, were co-transfected into the yeast strain Y2H Gold and incubated on the double dropout (DDO) medium (SD/−Leu/-Trp) at 30 °C for 3 days. Subsequently, monoclonal colonies were spotted on QDO medium.

### GST pull-down assay

NlTwd, NlToEVE14, and NlToEVE14-N mutants were expressed in prokaryotic and eukaryotic expression systems, respectively. For prokaryotic expression, the target sequences were cloned into PET-28a (Novagen, Darmstadt, Germany) for fusion expression with His-tag and transfected into *Escherichia coli* strain Transetta (#CD801-02, Trans-Gen Biotech, Beijing, China). Protein expression was induced by adding 0.1 mM isopropyl β-D-thiogalactoside (IPTG, #A100487, Sango Biotechnology) at 28 °C for 6 h. For eukaryotic expression, the target sequences were cloned into the PX3-FLAG-PCDNA vector (Sigma-Aldrich) for fusion expression with flag-tag and transfected into Human embryonic kidney (HEK) 293 T cells (ATCC, CRL-3216). The 293 T cells were maintained in Dulbecco's Modified Eagles Medium (DMEM, #2317091, VivaCell, Shanghai, China) that was supplemented with 10% fetal bovine serum (FBS, #F8318, Gibco, New York, USA), penicillin (100 U/ml), and streptomycin (100 U/ml) at 37% in a humidified incubator that contained 5% $CO_2$. The cells were collected 36 h after transfection. The expression of recombinant proteins was detected by Western blot assay using the His-tag antibodies (1:10,000 dilution, #MA1-21315, ThermoFisher Scientific), Flag-tag antibodies (1:10,000 dilution, #MA1-91878, ThermoFisher Scientific), and horse-radish peroxidase-conjugated goat anti-mouse IgG antibody (1:10,000, #31430, Thermo Fisher Scientific) as described above. The results indicated that NlToEVE14 cannot be expressed in either of the expression systems. NlTwd was only expressed in *E. coli*, while NlToEVE14-C was only expressed in 293 T cells.

Subsequently, GST pull-down assay was conducted as previously described[90]. Briefly, the GST-NlTwd and GST proteins were incubated with glutathione-sepharose beads (#C600031-0005, Sango, Shanghai, China) at 4 °C for 2 h. After washing with PBST (consisting of PBS and 0.1% Triton-100, #A110694, Sango Biotechnology) for 4 times, the beads were blocked with 10% FBS for 1 h. Then, NlToEVE14-C-flag was loaded onto the beads and incubated at 4 °C overnight. The beads were further washed with PBST for 4 times, and the precipitate was mixed with protein loading buffer (#P1041, Solarbio, Beijing, China). The Western blot assay was performed to detect recombinant proteins using the His-tag or flag-tag antibodies.

### Transcripts profiles of NlTwd and effects of NlTwd knockdown on the biological properties of N. lugens

The development and tissue expression profiles of *NlTwd* in *N. lugens* were investigated using the same methods as described for *NlToEVE14*. To further explore the biological properties of *N. lugens* affected by NlTwd knockdown, ds*NlTwd* was synthesized and RNAi experiments were performed using the same method as NlToEVE14 knockdown. The impact of NlTwd knockdown on the fecundity of *N. lugens* was also conducted similarly as described above. In addition, the survival rate of *N. lugens* was recorded 10 days after ds*NlTwd* injection into the newly emerged female adults, and ds*GFP* was used as control.

### Analysis of DEGs after NlToEVE14 knockout

Transcriptomic analysis was performed to analyze DEGs between WT and KO-M1 strains. Considering the periodic expression of NlToEVE14 during nymph stages (Fig. 4; Supplementary Fig. 4a), a prolonged nymphal development after NlToEVE14 knockout/silencing (Fig. 5b; Supplementary Fig. 8b; Supplementary Fig. 9a), and a potential role of NlToEVE14 in insect molting (Supplementary Fig. 10). The 5[th] instar nymphs of *N. lugens*, 72 h after molting, were selected for this analysis. The insect samples were collected and homogenized using the TRIzol Total RNA Isolation Kit (#9109, Takara, Dalian, China). Total RNA was extracted following the manufacturer's protocols. The RNA samples were sent to Novogene Institute (Novogene, Beijing, China) for transcriptomic sequencing as described above.

Subsequently, the raw reads were filtered and the clean reads from each transcriptome were aligned to the reference genome sequences of *N. lugens* using HISAT2 v2.1.0[91]. Low-quality alignments were filtered using Sequence Alignment/Map tools (SAMtools) v1.7[92]. Transcripts per million (TPM) expression values were calculated using Cufflink v2.2.1[93]. The DEseq2 v2.2.1[94] was used to analyze the DEGs, and genes with a log2-ratio >1 and adjusted *p* value < 0.05 were selected.

GO enrichment analyses were performed using TBtools (version 1.0697)[95], and enriched *P*-values were calculated using the hypergeometric test: $P = 1 - \sum_{i=0}^{m-1} \left( \frac{\binom{M}{i}\binom{N-M}{n-i}}{\binom{N}{n}} \right)$, where N represents the number of genes with GO annotation, n represents the number of DEGs in N, M represents the number of genes in each GO term, and m represents the number of DEGs in each GO term[95].

### Screening and analysis of ToEVEs in arthropod genomes

To gain insight into potential integrations of ToEVEs in arthropods, all available genomes of arthropod species (1188 genomes in total) were downloaded from the NCBI genome database (https://www.ncbi.nlm.nih.gov/genome). Preliminary screening of potential ToEVEs in arthropod genomes was conducted by tBlastN using the same method described above, and consecutive ToEVEs within the host genomes were merged (e-value < 1 × 10⁻⁵). Information on the screened ToEVEs and the 1188 arthropod genomes is provided in Supplemental Data 1 and Supplemental Data 2.

To further explore the association of ToEVE-containing arthropods with the hosts of their cognate exogenous viruses, representative species in the classes Insecta, Arachnida, and Malacostraca were selected based on the number of available genomes and initially screened ToEVEs. ToEVEs were determined with more stringent criteria (e-value < 1 × 10⁻¹⁰ and a minimum length of 350 bp) and were subsequently extracted from the corresponding genomes, which were

further verified by a reciprocal BLAST search as described above (sequences of the identified ToEVEs are provided in Source Data). The extracted ToEVEs were then searched (BlastX) against the protein sequences of all available totiviruses to obtain the best-hit homology totivirus and its host species (provided in Source Data). Moreover, to determine the potential transcripts of these ToEVEs, corresponding transcriptomes of the species (containing at least one ToEVE) were retrieved from the NCBI SRA database (retrieved on 18[th], December 2022). Potential ToEVE transcripts were identified and analyzed from de novo reassembled transcriptomes by locally searching (BlastN) as described above. Supplemental Table 5 provides related information on the arthropod transcriptomes, and the potential transcripts of the identified ToEVEs are listed in corresponding Source Data.

### Statistics and reproducibility

Two-tailed unpaired Student's *t* test was used to analyze the results of developmental duration, adult longevity, female-male ratio, fecundity, and short-winged/long-winged analysis. The log-rank test (SPSS Statistics 19, Chicago, IL, USA) was applied to determine the statistical significance of survival distributions. The exact *P* value of each statistical test was provided in the figures and Source data file. No statistical method was used to predetermine sample size. No data were excluded from the analyses, except for the sterile female in the fecundity analyses. All samples were allocated randomly into experimental groups. All investigation were blinded to group allocation during data collection and analysis.

### Reporting summary

Further information on research design is available in the Nature Portfolio Reporting Summary linked to this article.

## Data availability

The RNA-seq data used for totivirus identification in *Nilaparvata lugens* and *Sogatella furcifera* have been deposited in the NCBI Sequence Read Archive (SRA) under accession number SRR19073262 and SRR11729951, respectively. Sequence data can be found in GenBank under the following accession numbers: Nilaparvata lugens toti-like virus 1, ON402804; Sogatella furcifera toti-like virus 1, ON402805; Sogatella furcifera toti-like virus 2, ON402806; Fushun totivirus 2, MZ210014; Laodelphax striatellus iflavirus 1, MG815140.1; Nilaparvata lugens honeydew virus-1, NC_038302.1; Nilaparvata lugens honeydew virus-2, NC_021566.1; Nilaparvata lugens honeydew virus-3, NC_021567.1, Nilaparvata lugens reovirus, GCF_000852065.1, Sogatella furcifera honeydew virus 1, MG818986.1, Sogatella furcifera totivirus 1, NC_040633.1, and Sogatella furcifera totivirus 2, NC_040704.1; piggyBac transposable element-derived protein 3-like, XP_039275920.1. The sRNA-seq raw data of *N. lugens*, *Laodelphax striatellus* and *S. furcifera* can be found in NCBI SRA under accession number: PRJNA834899, PRJNA834958, and PRJNA834900, respectively. The RNA-seq raw data used for the analysis of DEGs between *N. lugens* strains of WT and KO-M1 can be found in NCBI SRA under accession number PRJNA987576. The genomes of *N. lugens*, *L. striatellus*, and *S. furcifera*, were retrieved from NCBI genome database with the accession numbers GCA_014356525, GCA_014465815, and GCA_014356515.1, respectively. The LC-MS/MS-based proteomic data are available in ProteomeXchange under the accession numbers: PXD023965, PXD036431, PXD043983, and PXD044065. The protein data of three planthoppers can be found in InsectBase 2.0 under the following ID: *N. lugens*, IBG_00572, *L. striatellus*, IBG_00477 and *S. furcifera*, IBG_00709. The NCBI SRA accession numbers for the transcriptome raw data used in the expression analysis of planthoppers and representative arthropods are provided in the Supplemental Table 3 and Supplemental Table 5, respectively. The NCBI genome accessions for the representative arthropods are provided in the Supplemental Table 4. The authors declare that the data supporting the findings of this study are available

in the paper, Supplemental Table 1-6, and Supplemental Data 1-2. Source data are provided with this paper.

## Code availability

The codes used in mapping, depth estimation and polymorphism level analysis have been deposited in Github: https://github.com/lyy005/TotiEVEs.

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

## Acknowledgements

We thank Professor Mike J. Adams (Minehead, UK) and Mang Shi (Sun Yat-Sen University, China) for their valuable and constructive suggestions for improving the manuscript. This work was supported by the National Key Research and Development Plan in the 14th five-year plan (2021YFD1401100: H.J.H. and C.X.Z.), the National Natural Science Foundation of China (U20A2036: J.P.C and J.M.L.; 32270146: J.M.L.).

## Author contributions

J.L., C.Z., and J.C. conceived and designed the research. J.L., H.H., Y.L., Z.Y., Q.H., Y.Q., and Z.X. performed computational analyses. H.H., L.L., Y.H., Y.Z., T.L., G.L., Q.M., J.Z., and J.L. performed the experiments. J.L., H.H., Y.L., Z.Y. Z.S., and F.Y. interpreted results. J.L., H.H., C.Z., and J.C. wrote the paper. All authors reviewed and edited the paper.

## Competing interests

The authors declare no competing interests.
