## [Peer Review File · Nature Communications]

Co-option of a non-retroviral endogenous viral element in planthoppersREVIEWER COMMENTS

Reviewer #1 (Remarks to the Author):

The Ms "Novel cellular functions of arthropod genes recruited from domesticated non-retroviral integrated RNA virus sequences" presents results concerning the identification and characterization of endogenous-viral elements from totiviruses in the genomes and transcriptomes of three planthoppers. Additionally, the CRISPR/Cas9 approach was used to generate a mutation in one sequence, namely NIETLVE14, followed by fitness experiments, which indirectly suggest a role for NIETVLE14 on *Nilaparvata lugens*. However, there are no results showing that the mutations (4bp deletion or 8 bp insertion) introduced in NIETVLE14 result in no protein production and there is no explanation, or hypothesis, regarding the role of NIETVLE14 on *N. lugens* biology. This is concerning given that fitness analyses showed significant differences in the two mutant strains vs the wild type in nymph development, adult longevity and fecundity, so broad phenotypes. RNAi was also employed to reduce NIETVLE14 transcript levels, followed by fitness assessment. However, RNAi knockdown effect is not shown and again fitness assessment does not identify a clear cause-effect between NIETVLE14 knock-down and a fitness parameter. For readers who are not experts of planthoppers is also unclear which is the phylogenetic relationship among the three studied species (which could also help dating some of the identified viral integrations based on whether they are orthologous) and the reasons by viral integrations from totiviruses (and not other viruses) are relevant for planthoppers.

As authors indicate, studies on the biology of viral integrations from non-retroviral RNA viruses are still limited with respect to studies on ERVs, resulting in issues with the nomenclature. Most recent papers tend to use the acronym nrEVEs instead of NIRVs to refer to viral integrations from non-retroviral RNA viruses. I suggest authors to switch to nrEVEs.

Further detailed concerns:

Lines 56-67. In most animals, nrEVEs tend to be uncomplete ORFs which produce piRNAs. Few nrEVEs which are transcribed have been identified, but whether they are translated has not been defined yet. I think authors should explain the background a bit better, also to highlight the novelty of their work, which to me is that they were able to identify a protein from an nrEVE.

Line56. Please provide a reference. To my knowledge, there are only four reports of viral integrations from nonretroviral RNA viruses and their role in immunity: 1-Maori et al., 2007, which does not explain the mechanist role of the viral integration in immunity to cognate viral infection; 2-Fujino et al. 2014 which is described in the text; 3- Suzuki et al., 2020 which shows that a viral integration among the hundreds found in the *Aedes aegypti* genome contributes in acquiring tolerance (and not resistance) to cognate viral infection in a tissue-specific manner and 4-Tassetto et al 2019, who worked with cells.

Line 88. Please provide references supporting this statement.

Line 90. In most insects, viral integrations are not randomly scattered across the genome, but prevalent in piRNA clusters. Please revise.

Line 93-94. Please define what you mean as "mounting compelling indirect information". Do you mean transcriptome data or other? Please note precursors of piRNAs are mRNA trascripts, so observing transcription from a viral integration does not exclude piRNA production vs translation
Line104-107. Please provide the reason why studying totivirus integrations (and not integrations from other viruses) is significant in the context of planthoppers.

Lines 115-121. Why was metatranscriptomic analyses limited to totiviruses? .

Line 137. Are any of the identified ETLVEs orthologues between *S. furcifera*, *L. striatellus* and/or *N. lugens* given that most are derived from NIToLVI?

Line142-145. Why is it unexpected that almost all ETLVEs are more similar to NIToLV1 than any other totivirus?

Line158. provide a reference to support your statement.

Line 146-164. The word "some" is not ok in a scientific publication. You cannot say "some " of the NIETLVEs, you need to specify which NIETLVEs. Please revise throughtout this paragraph.

Line 172-175. This sentence is unclear to me. Are authors talking about orthologues integrations? How distant are *S. furcifera*, *L. striatellus* and *N. lugens* phylogenetically?

Line 180-184. How were ETLVEs that are similar in sequence distinguished? Additionally, when you tested for the occurrence of the viral integrations in individuals genomes based on read coverage,

which strategy did you use to make sure that you were actually looking at the viral integration you annotated and not rearrangements or different viral integrations, which are unique of the samples you analyzed?

Line 190-192. The three NIETLVEs on chr2 of *N. lugens* are separated encompass a region of 2080732 bp and are separated by 2000 or more than 7500 bp. What is in annotated in between them? I mean they can derive from the same integration or not, you do not have any data to support one or the other event.

Line 187. Why samples from Australia are so different? Any biological reasons related to the distribution of *N. lugens*? Also in the PCA (fig. S3 *Nilaparvata muiri* is also shown). Why? Are the viral integrations all orthologues between the two species? Because the most important part of the ms regards NIETVLE14, please highlight the distribution of this viral integration across populations.

Line 194-195. Please provide data on the phylogenetic diverge between *N. muiri* and *N. lugens*.

Line 198-214. this is very confusing to me. Please provide a justification for the strategy used to define slow and fast evolving genes. Additionally, associating heterozygosity with fast and slow evolving genes is quite confusing. They are too different concepts. If integrations are old and have not reached fixation yet, finding them in heterozygosity would suggest they are evolving under drift or negative selection. Please elaborate on which evolutionary forces you refer to? Selection (negative, positive, balancing; drift, ...)

Line 234. There are ad hoc tests to verify positive selection (see Derbyshire 2020 *Front Microbiology*, fmicb.2020.00644). Please provide support for your statement.

Line 258-265. Figure 3 does not show introns within viral integrations, but it shows that NIETLVE13, NIETLVE14 and LsETLVE1 may be part (an exon) of a coding sequence. NIETLVE12, 5 and 17 and LsETLVE2,4,6 and 9 show no introns and the structure of the transcript is unclear, possible it is a piRNA precursor given piRNAs were detected? "No conserved domains were identified in these ORF", what does it mean? ETLVE were identified because they have homology to the CP of the RdRP of totiviruses! You mean you did not find additional domains apart from the viral motives?

Line 280. Figure S5 shows a band of 700KDa which is defined as non-specific band. How did you base your statement? Did you cut and sequence this band? Please specify if the peptide corresponding to NIETLVE14 was found in all analyzed databases.

Line 297-308. Why two different wild types for the two mutants? Did you use two different strains in the CRISPR/cas9 experiment? If you compare the two mutants (KO-M1 and KO-M2) you should not see any significant difference in the fitness parameters, is it the case? In the fertility, please exclude sterile females to calculate the mean.

Line 337-338. What do you mean by "quality of the insect"?

Line341-342. I respectfully disagree, in *Aedes* mosquitoes, most nrEVE do not derive from flaviviruses, but from rhabdoviruses and chuviridae.

Line 356. Which subsequent analyses? Did you search transcriptomes or RNAseq datasets or other?

Line371. Please define better. Why do you say NIETLVE14 has co-evolved with its host? You should a correlation between mutations in NIETLVE14 and different phenotypes, from these data it is impossible to define a specific function for your protein.

Line 402-404. Please see note for lines 258-265.

Line432-436. Please elaborate on which basis you think ETLVEs contribute to host immunity. You have no results on this subject.

Line 471: please explain the origin of the planthopper strains you used.

Line 482-483: RNA was pooled for the 20 individuals or analyzed individually?

Line600-608. Please specify the developmental stage of the samples used for RNA extraction as small RNA production may be developmental and tissue specific.

Reviewer #2 (Remarks to the Author):

General comments:

The authors describe a major work on endogenous toti-like viral elements (ETLVEs) in arthropod genomes. NIETLVE14, an ETLVE in *Nilaparvata lugens*, has been co-opted by its host and plays essential roles in planthopper development and fecundity. Numerous NIRVS were identified in

arthropods and they may play significant roles in the functional diversity of arthropod genes. The wide array of methods used to characterize the ETLVEs integrated into the arthropod genomes is impressive. Overall, the manuscript reads well, the methods are solid, and the results are well laid out. I list here a main issue that I had and have attached line by line comments. The identification in Figures is easy to confuse, and many images have both lowercase and uppercase letters. For examples, Fig.3B-D shown in Line 271, while Fig.3B-ab, D-a shown in Line 273.

Minor comments:

Line 80 Please change to an immune-related gene, COMMD3, Need to add a comma.

Line 85 bornaviral N? What does 'N' represent?

Line 159 Please change to SfETLVEs

Line 269 Briefly outline whether the analysis of small RNA comes from their own sequencing or from public databases.

Line 271 Please delete 'production'

Line 287 Please add a space between '24-48' and 'h'.

Line 334 Please change to 'with the large numbers'

Line 483 Briefly described how libraries were constructed and provide a method citation.

Line 269 and 599 delete small RNA (sRNA), because this abbreviated description has already appeared on line 269.

Line 634 Please add a space between '48' and 'h'.

Line 658 with change to using.

Reviewer #3 (Remarks to the Author):

Its great science, excellent paper with new information. The authors identified and analyzed endogenous toti-like viral elements (ETLVEs) in three rice planthoppers, including *Nilaparvata lugens* (BPH), *Laodelphax striatellus* (SBPH), and *Sogatella furcifera* (WBPH). ETLVEs are integrated into the genomes of three planthopper species, with highly variable distribution and heterozygosity in planthopper populations. They also studied NIETLVE14, one of the ETLVEs in *N. lugens*, could be transcribed and translated into a functional protein related to BPH development and fecundity by CRISPR/Cas9 and RNAi technology, to demonstrate the successful recruitment of a novel cellular function for arthropods from a tamed NIRVS as the result of long-term host-virus co-evolution. To fully elucidate the story, it is better to provide more evidence.

1 There are more ETLVEs identified in *N. lugens* than *L. striatellus* and *S. furcifera*. Is it because there are less samples (populations) tested in *L. striatellus* and *S. furcifera*? And similar question on the

2 I noticed there are difference in ETLVEs among the 6 populations of *N. lugens*. Is there difference among the different migrating area? And the biological function on development and fecundity effected by NIETLVE14, which found in all the populations, was studied. Is it possible that there is specific function effected by the different ETLVEs in the different migrating area?

3 NIRVS are known to be involved in host antiviral immunity, and transmitting virus disease is the important measure of rice planthopper to threaten rice. Do NIRVS effect the virus transmitting ability of rice planthopper? Such as the RSV transmitted by SBPH, or RRSV transmitted by BPH.

4 The cellular function of NIETLVE14 was studied through knockout it by CRISPR/Cas9, the evidence will be more powerful if the prolonged development and decreasing hatching rate in mutation strain could be recovered by knock in NIETLVE14.

5 NIETLVE14 was relatively high abundance in the carcass and wing buds at both transcription and protein levels. Do the authors observe the wing development in the NIETLVE14 mutant strain? Is the development postponed related with the regulating of JH or MH?

Reviewer #1 (Remarks to the Author):

The Ms “Novel cellular functions of arthropod genes recruited from domesticated non-retroviral integrated RNA virus sequences” presents results concerning the identification and characterization of endogenous-viral elements from totiviruses in the genomes and transcriptomes of three planthoppers. Additionally, the CRISPR/Cas9 approach was used to generate a mutation in one sequence, namely NIETLVE14, followed by fitness experiments, which indirectly suggest a role for NIETVLE14 on *Nilaparvata lugens*.¹ However, there are no results showing that the mutations (4bp deletion or 8 bp insertion) introduced in NIETVLE14 result in no protein production and there is no explanation, or hypothesis, regarding the role of NIETVLE14 on *N. lugens* biology.² This is concerning given that fitness analyses showed significant differences in the two mutant strains vs the wild type in nymph development, adult longevity and fecundity, so broad phenotypes. RNAi was also employed to reduce NIETVLE14 transcript levels, followed by fitness assessment.³ However, RNAi knockdown effect is not shown and again fitness assessment does not identify a clear cause-effect between NIETVLE14 knock-down and a fitness parameter.⁴ For readers who are not experts of planthoppers is also unclear which is the phylogenetic relationship among the three studied species (which could also help dating some of the identified viral integrations based on whether they are orthologous) and⁵ the reasons by viral integrations from totiviruses (and not other viruses) are relevant for planthoppers.⁶ As authors indicate, studies on the biology of viral integrations from non-retroviral RNA viruses are still limited with respect to studies on ERVs, resulting in issues with the nomenclature. Most recent papers tend to use the acronym nrVEEs instead of NIRVs to refer to viral integrations from non-retroviral RNA viruses. I suggest authors to switch to nrVEEs.

Response: We greatly appreciate your valuable comments and constructive suggestion to our work.

¹To confirm the absence of NIETVLE14 in the two knockout planthopper strains, we have conducted the protein detection and no band corresponding to NIETVLE14 was observed in the mutant strains. Please refer to the newly provided Supplemental Figure S6. The specificity of the antibody for NIETVLE14 detection was explained in “Response 20” below.

²For the biological role of NIETVLE14, we agree that the fitness experiments and various phenotypes between WT and mutant strains only provide indirectly evidence for NIETVLE14 function. A reasonable hypothesis is that NIETVLE14 might play multiple biological roles related to the development and fecundity of *N. lugens* based on the phenotypes of mutant strains. Although it’s still currently unclear why knockout this gene results the observed phenotypes, we conducted transcriptomes analysis to determine differentially expressed genes (DEGs) between NIETLVE14 knockout mutant (KO-M1) and the control (WT). Our results indicated that some genes associated with development and reproductions are

significantly enriched (**Supplemental Figure S11**), which might partially explain the changed phenotypes observed in KO-M1 strain. Please also refer to our “Response 25” below.

³For the RNAi knockdown efficiency, we have provided the original data of qPCR in source data for Fig.4g-h, Supplemental Figure S5b, Supplemental Figure S9, and Supplemental Figure S10e-g. Please refer to the “**Source data.xlsx**” of the revised version.

⁴For the phylogenetic relationship among the three studied species, previous evolutionary study has indicated that the three planthopper (*N. lugens*, *L. striatellus*, and *S. furcifera*) are belonging to the same family Delphacidae. The species *S. furcifera* diverged from *L. striatellus* approximately 46.1 million years ago, whereas the common ancestor of these two planthopper species separated from *N. lugens* approximately 64.4 million years ago (Ma et al., 2021, 10.1111/1755-0998.13242). Please refer to our “Response 12” below.

⁵For the reason to choose totivirus for the study of viral integrations in planthoppers, please refer to our “Response 6 and 7”.

⁶For the abbreviation of “non-retroviral RNA viruses”, as suggested, we have changed “NIRVs” to “nrEVEs” and “ETLVEs” to “ToEVEs” throughout the ms, as well as all of the figures and Supplemental materials. However, for the consistent with the previous content of the ms, in this point-by-point response, we still use “NIRVs” and “ETLVEs” to avoid confusion.

All in all, the manuscript has been carefully modified and improved following your suggestions. Please find our detailed point-to-point responses to your comments below.

Further detailed concerns:

1. Lines 56-67. In most animals, nrEVEs tend to be uncomplete ORFs which produce piRNAs. Few nrEVEs which are transcribed have been identified, but whether they are translated has not been defined yet. I think authors should explain the background a bit better, also to highlight the novelty of their work, which to me is that they were able to identify a protein from an nrEVE.

Response 1: Thank you for your valuable suggestions. We agree and have added the related content as “Nevertheless, in most animals, the open reading frames (ORFs) of reported nrEVEs are disrupted, resulting in the generation of piRNAs. Only a limited number of nrEVEs with intact ORF have been detected to be transcribed, leaving uncertainties about whether these transcripts are further translated into functional protein” in lines 72-75 of the revised manuscript.

2. Line 56. Please provide a reference. To my knowledge, there are only four reports of viral integrations from nonretroviral RNA viruses and their role in immunity: 1-Maori et al., 2007, which does not explain the mechanist role of the viral integration in immunity to cognate viral infection; 2-Fujino et al. 2014 which is described in the text; 3- Suzuki et al., 2020 which shows that a viral integration among the hundreds found in the Aedes aegypti genome contributes in acquiring tolerance (and not resistance) to cognate viral infection in a tissue-specific manner and 4-Tassetto et al 2019, who worked with cells.

Response 2: Thank you for your suggestions. Two review articles “Blair et al., 2020; Frank and Feschotte, 2017” have been added to address the antiviral role of ERV and nrEVE in eukaryotes in lines 60-61 of the revised ms. The four reports for the functional roles of NIRVs in immunity have been described in lines 64-72 in the revised ms.

3. Line 88. Please provide references supporting this statement.

Response 3: The references “Crava et al., 2021; Palatini et al., 2022” related to the content have been added as suggested. Please see line 98 in the revised ms.

4. Line 90. In most insects, viral integrations are not randomly scattered across the genome, but prevalent in piRNA clusters. Please revise.

Response 4: Thank you for your suggestions. This sentence has been modified as “Although it has been suggested that nrEVEs are commonly found in piRNA clusters, these findings suggest that some of the nrEVEs might undergo protein-level domestication and have biological functions”. Please refer to lines 98-101 in the revised ms.

5. Line 93-94. Please define what you mean as “mounting compelling indirect information”. Do you mean transcriptome data or other? Please note precursors of piRNAs are mRNA trascripts, so observing transcription from a viral integration does not exclude piRNA production vs translation

Response 5: Thank you for your comments and we are sorry for the ambiguous statement. We have deleted “despite the mounting compelling indirect information provided by genomics studies” to avoid confusion (lines 104-105).

6. Line104-107. Please provide the reason why studying totivirus integrations (and not integrations from other viruses) is significant in the context of planthoppers.

Response 6: Thank you for your valuable comments. Totiviruses have been selected for the study of nrEVEs of planthopper mainly due to the following reasons. The main reason is that it has been commonly observed that species with EVEs are often closely related to the host of EVE-homology exogenous viruses; a typical example is mosquito flavivirus-derived EVEs with mosquito-infecting flaviviruses. Hence, we concentrated on the exogenous viruses that can infect the three planthoppers, conducting a preliminary screen (tBlastN) against the host genomes using a collection of these viruses. The results indicated that only totiviruses are integrated into the planthopper genomes with significant hits (e-value $<1\times 10^{-10}$). Furthermore, viruses in the family *Totiviridae* have widely reported to be integrated into various eukaryotic genomes (Taylor and Bruenn, 2009; Aiewsakun and Katzourakis, 2015; Flynn and Moreau, 2019; Russo et al., 2019). Considering these, totiviruses were finally chosen for planthopper nrEVE analysis in this study. We have added sentence “Considering the novel totiviruses recently discovered in planthoppers, it is intriguing to investigate whether totiviruses are also integrated into genomes of the three planthoppers.” in lines 116-118 of the revised ms. And the preliminary screen for the presence of nrEVEs was added in the section “Material and Methods” in lines 580-592 of the revised ms.

References

Aiewsakun, P., and Katzourakis, A. (2015). Endogenous viruses: Connecting recent and ancient viral evolution. *Virology* 479-480, 26-37. 10.1016/j.virol.2015.02.011.

Flynn, P.J., and Moreau, C.S. (2019). Assessing the Diversity of Endogenous Viruses Throughout Ant Genomes. *Front. Microbiol.* 10, 1139. 10.3389/fmicb.2019.01139.

Russo, A.G., Kelly, A.G., Enosi Tuipulotu, D., Tanaka, M.M., and White, P.A. (2019). Novel insights into endogenous RNA viral elements in *Ixodes scapularis* and other arbovirus vector genomes. *Virus Evol.* 5, vez010. 10.1093/ve/vez010.

Taylor, D.J., and Bruenn, J. (2009). The evolution of novel fungal genes from non-retroviral RNA viruses. *BMC Biol.* 7, 88. 10.1186/1741-7007-7-88.

7. Lines 115-121. Why was metatranscriptomic analyses limited to totiviruses?

Response 7: Please refer to our response of the previous question (Response 6). Because we subsequently focused on the analysis of totiviruses, so the metatranscriptomic analyses of exogenous viruses was primarily limited to totiviruses. We’ve modified the sentence for the clarity. Please refer to lines 128-135 in the revised ms.

8. Line 137. Are any of the identified ETLVEs orthologues between *S. furcifera*, *L. striatellus* and/or *N. lugens* given that most are derived from NIToLVI?

Response 8: Thank you for the valuable suggestion. We have conducted orthologous analysis based on the predicted proteins of the three planthoppers (including ETLVEs). The results indicated that ETLVEs can be assigned to three orthologous groups.

Group-1: NIETLVE1, NIETLVE6, NIETLVE7, NIETLVE8, NIETLVE18, LsETLVE1, SfETLVE1.

Group-2: NIETLVE11, NIETLVE15.

Group-3: NIETLVE13, NIETLVE14.

Please refer to newly provided Supplemental Table S2 and lines 196-200, 659-666 in the revised ms.

9. Line142-145. Why is it unexpected that almost all ETLVEs are more similar to NIToLV1 than any other totivirus?

Response 9: Thank you for your comments and we are sorry for the confusion. One possible reason for this scenario might be because that NIToLV1 is an insect specific virus of *N. lugens*, which has more chance to be integrated into the host genome. However, we haven't identified SfToLV1- and SfToLV2-derived nrEVEs although *S. furcifera* is infected by these two viruses. So we do agree that the word "Unexpectedly" is improperly used and it has been removed in lines 162 of the revised ms.

10. Line158. provide a reference to support your statement.

Response 10: The reference (Zhang et al., 2018) was added accordingly. Please refer to lines 179 in the revised ms.

11. Line 146-164. The word "some" is not ok in a scientific publication. You cannot say "some" of the NIETLVEs, you need to specify which NIETLVEs. Please revise throughout this paragraph.

Response 11: Thank you for your suggestion and the related sentences has been modified accordingly throughout this paragraph. Please refer to lines 166-170, 184-188 in the revised ms.

12. Line 172-175. This sentence is unclear to me. Are authors talking about orthologues integrations? How distant are *S. furcifera*, *L. striatellus* and *N. lugens* phylogenetically?

Response 12: We are sorry for the confusion. Yes, we are talking about orthologues of ETLVE integrations in planthopper genomes. Previous evolutionary study has indicated that the three planthopper (*N. lugens*, *L. striatellus*, and *S. furcifera*) used in this study are belonging to the same family Delphacidae. The species *S. furcifera* diverged from *L. striatellus* approximately 46.1 million years ago, whereas the common ancestor of these two planthopper species separated from *N. lugens* approximately 64.4 million years ago (Ma et al., 2021) (see the figure below). Thus, the wide integration of ETLVEs (orthologous group-1 and group-2) derived from totivirus that infecting *N. lugens* (NIToLV1) in the genomes of the three planthoppers suggested that the ancient viral integration events might have occurred before 64.4 million years ago. Alternatively, orthologous group-3 of ETLVEs (NIETLVE13 and NIETLVE14) was homologous to the totivirus of *S. furcifera* (SFToLV2), implying the possibility of another independent viral integration event in ancient times. Please refer to lines 200-214 in the revised ms.

Figure of Response 12 (Modified from Ma et al., 2021)

Reference:

Ma, W., Xu, L., Hua, H., Chen, M., Guo, M., He, K., Zhao, J., and Li, F. (2021). Chromosomal-level genomes of three rice planthoppers provide new insights into sex chromosome evolution. *Mol. Ecol. Resour.* 21, 226-237. [10.1111/1755-0998.13242](https://doi.org/10.1111/1755-0998.13242).

13. Line 180-184. How were ETLVEs that are similar in sequence distinguished? Additionally, when you tested for the occurrence of the viral integrations in individuals genomes based on read

Editorial Note: Figure above adapted from Ma, W, Xu, L, Hua, H, et al. Chromosomal-level genomes of three rice planthoppers provide new insights into sex chromosome evolution. *Mol Ecol Resour.* 2021; 21: 226–237. <https://doi.org/10.1111/1755-0998.13242>, with permission. © 2020 John Wiley & Sons Ltd

coverage, which strategy did you use to make sure that you were actually looking at the viral integration you annotated and not rearrangements or different viral integrations, which are unique of the samples you analyzed?

Response 13: Thank you for your comments. We have used EMBOSS distmat (<https://www.bioinformatics.nl/cgi-bin/emboss/distmat>) to calculate the similarity among the three groups of these ETLVEs. The highest identity was found to be 76.15% between NIETLVE1 and NIETLVE3 in group-1, while group-2 and group-3 exhibited identities of 74.01% and 72.62%, respectively. These differences can be easily distinguished when mapping the ETLVEs to the individual planthopper genome using BWA MEM version 0.7.17.

Moreover, we used the strategy considering both the read coverage and the genome sequencing depth of the ETLVEs. The presence of ETLVEs were considered in the genome only when the ETLVEs with sequencing depth > 5 and coverage >50% for *S. furcifera* or *L. striatellus*, and sequencing depth > 1 and coverage >50% for *N. lugens*. The relative information has been added in the section of “Material and Methods”. Please refer to lines **683-687** in the revised ms. Nevertheless, it is possible that genomes of some individual planthoppers might contain rearranged ETLVEs which is difficult to be detected with this strategy. Thus, our results present an overview of these ETLVEs in the genomes of planthopper populations as indicated in line **217** of the revised ms.

14. Line 190-192. The three NIETLVEs on chr2 of *N. lugens* are separated encompass a region of 2080732 bp and are separated by 2000 or more than 7500 bp. What is in annotated in between them? I mean they can derive from the same integration or not, you do not have any data to support one or the other event.

Response 14: Thank you for the suggestion. Considering the importance of transposable elements (TEs) for the generation of EVEs, the two sequences between NLETLVE3-5 were extracted and used for the identification of potential TEs with CENSOR (<https://www.girinst.org/censor/index.php>). Interestingly, a piggyBac Transposon (piggyBac transposable element-derived protein 3-like) was predicted between NLETLVE3 and NLETLVE4, suggesting that NLETLVE3 and NLETLVE4 might originate from the same insertion event. Thus, we proposed that NLETLVE3-5 is possibly derived from the same viral integration events supported by the following evidences: 1, The NLETLVE3-5 are adjacent in Chr2 of the genome (Fig.1a); 2, The NLETLVE3-5 exhibit similar presence pattern in the individual genomes of planthopper population, which can be distinguished from NLETLVE2 (Fig.2a); 3, A piggyBac Transposon was identified between NLETLVE3 and NLETLVE4. Please refer to lines **234-238 in the revised ms.**

15. Line 187. Why samples from Australia are so different? Any biological reasons related to the distribution of *N. lugens*? Also in the PCA (fig. S3 *Nilaparvata muiri* is also shown). Why? Are the viral integrations all orthologues between the two species? Because the most important part of the ms regards NIETVLE14, please highlight the distribution of this viral integration across populations.

Response 15: Thank you for your valuable suggestion. It's true that NIETLVEs of Australia population are very different with other populations. Our previous works on the migratory route investigation of *N. lugens* based on the individual genome suggested that Australia population exhibit large genetic divergence and form a distinctive group compare to the other Asia populations, which might be explained by geographic barrier (Hu et al., 2019). We have added this information in lines 222-223, 227-232 of the revised ms.

For *Nilaparvata muiri*, considering that there are only two individual genomes available for this species, we have removed *N. muiri* and performed the PCA analysis again to improve the robustness of this result. Please see the modified Supplemental Figure S3 and line 695 in the revised ms. Since the reference genome of *N. muiri* is currently unavailable, we also tried to assemble the full genome of *N. muiri* using the previous sequenced raw reads. However, the assembled genome quality of *N. muiri* is very low, and we failed to assemble the complete sequences of neither NIETVLE13 nor NIETVLE14.

Reference:

Hu, Q.-I., Zhuo, J.-C., Ye, Y.-X., Li, D.-T., Lou, Y.-H., Zhang, X.-Y., Chen, X., Wang, S.-L., Wang, Z.-C., and Lu, J.-B. (2019). Whole genome sequencing of 358 brown planthoppers uncovers the landscape of their migration and dispersal worldwide. bioRxiv, 798876.

16. Line 194-195. Please provide data on the phylogenetic diverge between *N. muiri* and *N. lugens*.

Response 16: Thank you for your comments. As indicated in our "Response 12", molecular dating analysis showed that *N. lugens* diverged from the common ancestor of *L. striatellus* and *S. furcifera* approximately 64.4 million years ago. We proposed that the insertion event of NIETLVE13 and NIETLVE14 took place after this divergence, subsequently becoming stably inherited within the genome of the planthopper species in the genus *Nilaparvata*. Please refer to lines 243-246 in the revised ms.

17. Line 198-214. this is very confusing to me. Please provide a justification for the strategy used to define slow and fast evolving genes. Additionally, associating heterozygosity with fast and

slow evolving genes is quite confusing. They are too different concepts. If integrations are old and have not reached fixation yet, finding them in heterozygosity would suggest they are evolving under drift or negative selection. Please elaborate on which evolutionary forces you refer to? Selection (negative, positive, balancing; drift,)

Response 17: Thank you for this important comment and we are sorry for the confusion. As of now, the evolution of NIRVS in population level was rarely studied and mostly focused on arboviral vector *Aedes* spp. Thus, the strategy used to define slow and fast evolving genes are following the approach suggested by NIRVS studies in mosquito populations (Crava et al. 2021; Pischedda et al. 2019). The references were added accordingly in the revised ms (lines 708, 1150-1153). Briefly, the protein-coding genes of the three planthoppers were assigned into orthologous groups. Subsequently, the single-copy orthologs were then aligned and the pairwise p-distance of each ortholog alignment was calculated. Finally, the top 5% and the bottom 5% genes were selected as planthopper FEGs and SEGs, respectively. Please refer to lines 708-717 in the revised ms.

We do agree with the reviewer and we are sorry for the confusion to associate the concept of heterozygosity with fast and slow evolving genes. According to the previous evolution analysis of NIRVS in mosquito population (Crava et al. 2021; Pischedda et al. 2019), it is more appropriate to use “**polymorphism level**” instead of “heterozygosity”. So we modified the relevant text throughout the revised ms (as well as the figures) to avoid the confusion. For the types of evolutionary selection forces of ETLVEs, we think that it’s unreliable to draw conclusion based on the current available evidences. For example, the polymorphism levels of NIETLVE11-14 are similar and are comparable to the FEGs and SEGs of *N. lugens* (Fig.2b), implying that they might under similar selection force. However, NIETLVE11 is variably distributed in the populations of *N. lugens*, whereas NIETLVE12-14 is present in almost all of the individuals (fixed nrEVEs). Similar scenario was also observed for NIRVS in the populations of mosquito (Crava et al. 2021). We believe that new evidence will be revealed in the future to address this issue when more NIRVS studies are conducted at population level.

Reference:

Crava, C.M., Varghese, F.S., Pischedda, E., Halbach, R., Palatini, U., Marconcini, M., Gasmi, L., Redmond, S., Afrane, Y., Ayala, D., et al. (2021). Population genomics in the arboviral vector *Aedes aegypti* reveals the genomic architecture and evolution of endogenous viral elements. *Mol. Ecol.* 30, 1594-1611. 10.1111/mec.15798.

Pischedda, E., Scolari, F., Valerio, F., Carballar-Lejarazu, R., Catapano, P.L., Waterhouse, R.M., and Bonizzoni, M. (2019). Insights Into an Unexplored Component of the Mosquito Repeatome: Distribution and Variability of Viral Sequences Integrated Into the Genome of the Arboviral Vector *Aedes albopictus*. *Frontiers in genetics* 10, 93. 10.3389/fgene.2019.00093.

18. Line 234. There are ad hoc tests to verify positive selection (see Derbyshire 2020 Front Microbiology, fmicb.2020.00644). Please provide support for your statement.

Response 18: Thank you for your valuable suggestion. As referred in our previous response (Response 17), it might inappropriate to state that these ETLVEs are under positive selection presently. Moreover, VCFTools was chosen to calculate Tajima's D as suggested by Derbyshire et al. (2020) for the Asia populations of *N. lugens*, and the results showed various signals of selections. Please refer to the file "TajimaD analysis-for reviewer 1.xlsx". Therefore, the sentence was modified accordingly. Please see line 289 in the revised ms.

19. Line 258-265. Figure 3 does not show introns within viral integrations, but it shows that NIETLVE13, NIETLVE14 and LsETLVE1 may be part (an exon) of a coding sequence. NIETLVE12, 5 and 17 and LsETLVE2,4,6 and 9 show no introns and the structure of the transcript is unclear, possible it is a piRNA precursor given piRNAs were detected? "No conserved domains were identified in these ORF", what does it mean? ETLVE were identified because they have homology to the CP of the RdRP of totiviruses! You mean you did not find additional domains apart from the viral motives?

Response 19: Thank you for your comments and we are sorry for the incorrect description. Yes, none of the three ETLVEs (NIETLVE13, NIETLVE14, and LsETLVE1) were located in the introns and they presented as part of the exons. We have corrected the mistake and modified the sentences (lines 317-321 of the revised ms). We do agree that these ETLVEs with abundant piRNA production are possibly served as piRNA precursors (lines 335-336 of the revised ms). We also sorry for the inaccurate description related to the conserved domain predication. And that's true that we failed to find additional domains apart from the viral motifs. Please refer to lines 324-325 of the revised ms.

20. Line 280. Figure S5 shows a band of 700KDa which is defined as non-specific band. How did you base your statement? Did you cut and sequence this band? Please specify if the peptide corresponding to NIETLVE14 was found in all analyzed databases.

Response 20: Thank you for your comments. We do agree that the specific for the antibody is essential for this work. The specificity of this ~100kDa band, as well as the non-specificity of the ~70kDa band, is unequivocally supported by the following evidences.

- 1) The predicated molecular weight of NIETLVE14 is approximately 106 kDa that is consistent with the size of this ~100kDa band (Supplemental Figure S5c);
- 2) The ~70kDa band can still be easily detected when we knockdown of NIETLVE14 (treated with ds*NIETLVE14*) whereas the ~100kDa band obviously got weaker after dsRNA treatment (Supplemental Figure S5c); Please see the modified figure legend of Supplemental Figure S5 in lines **1324-1327** of the revised ms.
- 3) The specificity of the antibody was further confirmed using CRISPR/Cas9-mediated knockout approach. The ~100 kDa band disappeared in both of the two NIETLVE14-knockout mutant strains (KO-M1 and KO-M2) of planthoppers compare to the control (WT), whereas the ~70kDa band can still be readily detected in KO-M1 and KO-M2 strains. Please see the newly provided **Supplemental Figure S6** and lines **367-370**, **1333-1338** of the revised ms.

For the identification of peptide corresponding to NIETLVE14, we have analyzed three proteomic databases of *N. lugens* (ProteomeXchange accession: PXD036431, PXD043983, and PXD044065), and peptide of NIETLVE14 was identified in one database (PXD036431) which might due to the insensitive detection ability of the proteomic technology (compared to that of the transcriptome). Please refer to lines **343-345**, **768-769** of the revised ms.

21. Line 297-308. Why two different wild types for the two mutants? Did you use two different strains in the CRISPR/cas9 experiment? If you compare the two mutants (KO-M1 and KO-M2) you should not see any significant difference in the fitness parameters, is it the case? In the fertility, please exclude sterile females to calculate the mean.

Response 21: Thank you for your valuable comments. In our lab, only one *N. lugens* strain/population is maintained which was used for the CRISPR/cas9 experiments in this study. Actually, the knockout technology of *N. lugens* is still very challenging. We tried several times to conduct this experiment, and finally, two NIETLVE14-mutants were successfully generated and confirmed in two independent trials. Subsequently, we perform the bioassay and compared the two mutant populations (namely KO-M1 and KO-M2 in the ms) to their corresponding WT controls (namely WT1 and WT2 in the ms) separately. The **comparison group 1** (WT vs KO-M1) was provided in main Figure 4 and Supplementary Figure 7, and the **comparison group 2** (WT vs KO-M2) in Supplementary Figure 8. This treatment was also conducted in our previous publication in Nature Communication (Huang et al., 2023, 10.1038/s41467-023-36403-5). To avoid the confusion, we use “WT” instead of “WT1” and “WT2” throughout the revised ms (as well as all of the figures).

Reference

Huang, H.J., Wang, Y.Z., Li, L.L., Lu, H.B., Lu, J.B., Wang, X., Ye, Z.X., Zhang, Z.L., He, Y.J., Lu, G., et al. (2023). Planthopper salivary sheath protein LsSP1 contributes to manipulation of rice plant defenses. *Nat. Commun.* 14, 737. 10.1038/s41467-023-36403-5.

We have compared the fitness parameters of the two mutants (KO-M1 and KO-M2), and found they are very similar in comparison to the corresponding WT population as detailed below.

1) Total development duration period in comparison group 1 was extended for 1.49 days (***, $P < 0.001$), while in comparison group 2 was 0.81 days (***, $P < 0.001$).

2) Female longevity in comparison group 1 was decreased by 31.6% (***, $P < 0.001$), while in comparison group 2 was 31.4% (**, $P = 0.003$).

3) Male longevity in comparison group 1 was decreased by 17.2% (*, $P = 0.016$), while in comparison group 2 was 21.7% (*, $P = 0.024$).

4) Number of offspring in comparison group 1 was decreased by 54.3% (***, $P < 0.001$), while in comparison group 2 was 57.2% (***, $P < 0.001$).

5) Number of eggs in comparison group 1 was decreased by 40.6% (***, $P < 0.001$), while in comparison group 2 was 31.6% (**, $P = 0.004$).

6) Hatch rate in comparison group 1 was decreased by 19.8% (***, $P < 0.001$), while in comparison group 2 was 35.0% (***, $P < 0.001$).

We have provided the P value of each comparison in the revised Figures. Moreover, the sterile females were excluded and the data was reanalyzed as indicated in the revised figures and line 840 of the revised ms.

22. Line 337-338. What do you mean by “quality of the insect”?

Response 22: We are sorry for the confusion. Here we want to point out that both the genome size and the genome quality of the insects are essential for the discovered number of EVEs. Please refer to lines 426 in the revised ms.

23. Line 341-342. I respectfully disagree, in *Aedes* mosquitoes, most nrEVE do not derive from flaviviruses, but from rhabdoviruses and chuviridae.

Response 23: Thank you for pointing out the inapposite statement. We have modified the sentence as suggested and add one reference accordingly. Please refer to lines 431-432 and 1004-1006 in the revised ms.

24. Line 356. Which subsequent analyses? Did you search transcriptomes or RNAseq datasets or other?

Response 24: Thank you for your comments. Yes, we searched the de novo assembled transcriptomes (raw data retrieved from SRA database) using these ETLVEs to identify potentially transcribed ones. In the revised ms, we have added the relevant content accordingly (lines 448-449 of the revised ms). Detailed method for this process was indicated in the section of Material and Methods (lines 933-938 of the revised ms).

25. Line371. Please define better. Why do you say NIETLVE14 has co-evolved with its host? You should a correlation between mutations in NIETLVE14 and different phenotypes, from these data it is impossible to define a specific function for your protein.

Response 25: Thank you for your valuable comments and we are sorry for the inappropriate statement. We agree that the specific function of NIETLVE14 can't be defined according to the current evidences. We have modified the sentence to avoid misunderstanding. Please refer to lines 466 in the revised ms.

Moreover, to better understand the NIETLVE14 function, differentially expressed genes (DEGs) were determined between NIETLVE14 knockout mutant (KO-M1) and the control (WT). The results showed that DEGs are clearly associated with planthopper development, especially for the cuticle-associated genes (Supplemental Figure S11), which was also supported by our point-to-point Y2H assays (Supplemental Figure S10). In addition, GO analysis also indicated that reproduction-associated genes (reproductive behavior) are significantly enriched, which might partially explain the changed phenotype related to reproduction in KO-M1 strain. Nevertheless, the specific functions of NIETLVE14 still need further investigation. Please see the newly provided Supplemental Figure S11 and lines 404-417, 899-915 of the revised ms.

26. Line 402-404. Please see note for lines 258-265.

Response 26: Please refer to the Response 19. The sentence was modified to avoid the mistake. Please refer to lines 499-502 in the revised ms.

27. Line432-436. Please elaborate on which basis you think ETLVEs contribute to host immunity. You have no results on this subject.

Response 27: Sorry for the inappropriate statement. We have modified the sentence accordingly for the clarity. Please refer to lines 531-534 in the revised ms.

28. Line 471: please explain the origin of the planthopper strains you used.

Response 28: Done. The origin of the planthopper strains were provided as suggested. Please refer to lines 574-576 in the revised ms.

29. Line 482-483: RNA was pooled for the 20 individuals or analyzed individually?

Response 29: Sorry for the confusion. Yes, RNA was pooled for the 20 individuals. The sentence was modified for the clarity (lines 600-601 in the revised ms).

30. Line 600-608. Please specify the developmental stage of the samples used for RNA extraction as small RNA production may be developmental and tissue specific.

Response 30: Thank you for the suggestion. We agree that this information is important for small RNA analysis. In our study, a pool of approximately 20 ovaries (dissected from female insects) was used for this analysis. Please refer to lines 755-756 in the revised ms.

Reviewer #2 (Remarks to the Author):

General comments:

The authors describe a major work on endogenous toti-like viral elements (ETLVEs) in arthropod genomes. NIETLVE14, an ETLVE in *Nilaparvata lugens*, has been co-opted by its host and plays essential roles in planthopper development and fecundity. Numerous NIRVS were identified in arthropods and they may play significant roles in the functional diversity of arthropod genes. The wide array of methods used to characterize the ETLVEs integrated into the arthropod genomes is impressive. Overall, the manuscript reads well, the methods are solid, and the results are well laid out. I list here a main issue that I had and have attached line by line comments.

Response: Thank you for your positive comments.

The identification in Figures is easy to confuse, and many images have both lowercase and uppercase letters. For examples, Fig.3B-D shown in Line 271, while Fig.3B-ab, D-a shown in Line 273.

Response: We are sorry for the confusion on the mixed usage of lowercase and uppercase letter in the Figures. We have rearranged the letters combing a, b, c and i, ii, iii for all of the related Figures to avoid confusion. Please see the modified Figures in the revised version.

Minor comments:

1. Line 80 Please change to an immune-related gene, COMMD3, Need to add a comma.

Response 1: Done. Please refer to lines 89 in the revised ms.

2. Line 85 bornaviral N? What does 'N' represent?

Response 2: Sorry for the confusion. The 'N' represents nucleoprotein of bornavirus. Please refer to lines 94 in the revised ms.

3. Line 159 Please change to SfETLVEs

Response 3: Done. According to the suggestion of the other Reviewer, we have changed SfETLVEs to SfToEVEs throughout the ms. Please refer to lines 181 in the revised ms.

4. Line 269 Briefly outline whether the analysis of small RNA comes from their own sequencing or from public databases.

Response 4: Thank you for your comments. These small RNAs were derived from the dissected ovaries of our lab populations of the three planthopper species. Please refer to lines 332-333, 755-756 in the revised ms.

5. Line 271 Please delete 'production'

Response 5: Done. Please refer to lines 329 in the revised ms.

6. Line 287 Please add a space between '24-48' and 'h'.

Response 6: Done. Please refer to lines 354 in the revised ms.

7. Line 334 Please change to 'with the large numbers'

Response 7: Done. Please refer to lines 423 in the revised ms.

8. Line 483 Briefly described how libraries were constructed and provide a method citation.

Response 8: Thank you for your suggestion. We have provided detailed method for the construction of the libraries and provided the method citation. Please refer to lines 601-610, 1058-1060 in the revised ms.

9. Line 269 and 599 delete small RNA (sRNA), because this abbreviated description has already appeared on line 269.

Response 9: Done. Please see line 754 in the revised ms.

10. Line 634 Please add a space between '48' and 'h'.

Response 10: Done. Please see line 790 in the revised ms.

11. Line 658 with change to using.

Response 11: Done. Please see line 813-814 in the revised ms.

Reviewer #3 (Remarks to the Author):

Its great science, excellent paper with new information. The authors identified and analyzed endogenous toti-like viral elements (ETLVEs) in three rice planthoppers, including *Nilaparvata lugens* (BPH), *Laodelphax striatellus* (SBPH), and *Sogatella furcifera* (WBPH). ETLVEs are integrated into the genomes of three planthopper species, with highly variable distribution and heterozygosity in planthopper populations. They also studied NIETLVE14, one of the ETLVEs in *N. lugens*, could be transcribed and translated into a functional protein related to BPH development and fecundity by CRISPR/Cas9 and RNAi technology, to demonstrate the successful recruitment of a novel cellular function for arthropods from a tamed NIRVS as the result of long-term host-virus co-evolution. To fully elucidate the story, it is better to provide more evidence.

Response: We greatly appreciate your positive comments and valuable suggestions to our work. The manuscript has been carefully modified and improved following your suggestions. Please see our detailed responses below.

1. There are more ETLVEs identified in *N. lugens* than *L. striatellus* and *S. furcifera*. Is it because there are less samples (populations) tested in *L. striatellus* and *S. furcifera*? And similar question on the

Response 1: Thank you for your comments. In this study, the ETLVEs were identified using the previously reported genomes of three planthoppers (indicated in lines 633-636 of the revised ms), and only one population of each planthopper species was used for the genome assembly. Moreover, the patterns of ETLVEs presence in the genome are similar for the Asia populations of *N. lugens* (Fig.2A), suggesting that ETLVEs numbers might not correlated with insect populations. We speculate the main reason for more ETLVEs identified in *N. lugens* (compare to the other planthopper species) is the discovery of *Nilaparvata lugens* toti-like virus 1 (NIToLV1) that exhibits homologous to the majority of the identified ETLVEs (Fig.1A-C). Both previous reports and our large-scale investigations revealed that species with NIRVS are often closely related to the NIRVS-homology exogenous viruses infecting the same host species. And the discovery of specific exogenous viruses is crucial for the successful identification of NIRVS as indicated in lines 428-432, 444-455 of the revised ms and Fig.5B. And it can be expected that new ETLVEs will be revealed in planthopper genomes with the discovery of novel toti-like viruses infecting the three planthopper species in the future. Furthermore, the obvious larger genome size of *N. lugens* (1088M) compare to *L. striatellus* (540M) and *S. furcifera* (656) might also contribute to more ETLVEs identified in *N. lugens*.

2. I noticed there are difference in ETLVEs among the 6 populations of *N. lugens*. Is there difference among the different migrating area? And the biological function on development and fecundity effected by NIETLVE14, which found in all the populations, was studied. Is it possible that there is specific function effected by the different ETLVEs in the different migrating area?

Response 2: Thank you for your valuable comments. Yes, there are differences in ETLVEs among the 6 populations of *N. lugens*, especially between the Australia population and the 5 Asia populations. Our previous works on the migratory route investigation of *N. lugens* based on the individual genome suggested that Australia population exhibit large genetic divergence and form a distinctive group compare to the other Asian populations, which might be explained by geographic barrier (Hu et al., 2019). We have added this information in lines 222-223, 227-232 of the revised ms.

For NIETLVE14, it is presented in all of the screened individual genomes of *N. lugens* populations with high cover percentages, indicating that NIETLVE14 might be important for this species. Our bioinformatic analysis and subsequent knockout/knockdown experiments revealed that NIETLVE14 might have been domesticated as a novel functional protein crucial for *N. lugens* biology. Besides, we also noticed that some ETLVEs greatly varied across different populations, and we agree that there might be specific function for these ETLVEs in the different migrating area. Nevertheless, since that all these *N. lugens* individuals were collected from the field that were not maintained in the lab, we did not compare the biological function of the ETLVEs for the different populations. Nevertheless, it is intriguing to investigate the specific function of these varied ETLVEs in different insect population in the future study.

Reference:

Hu, Q.-I., Zhuo, J.-C., Ye, Y.-X., Li, D.-T., Lou, Y.-H., Zhang, X.-Y., Chen, X., Wang, S.-L., Wang, Z.-C., and Lu, J.-B. (2019). Whole genome sequencing of 358 brown planthoppers uncovers the landscape of their migration and dispersal worldwide. *bioRxiv*, 798876.

3. NIRVS are known to be involved in host antiviral immunity, and transmitting virus disease is the important measure of rice planthopper to threaten rice. Do NIRVS effect the virus transmitting ability of rice planthopper? Such as the RSV transmitted by SBPH, or RRSV transmitted by BPH.

Response 3: Thank you for your valuable suggestions. A notable example for antiviral immunity of NIRVS was demonstrated in mosquitoes that NIRVS-derived piRNAs can successfully control the replication of cognate viruses (Suzuki et al.,

2020). And we do agree that it's very interesting to evaluate whether planthopper NIRVS could affect the acquisition and transmission of rice viruses with the similar strategy. Actually, we failed to identify any NIRVS that exhibit homologous to economically important rice viruses transmitted by planthoppers. Moreover, we also failed to identify potential target sites in the genome of these rice viruses using these ETLVE-derived piRNAs. Nonetheless, it can't exclude that these piRNAs might affect the replication of exogenous viruses using more sophisticated way, for example, through targeting host antiviral-related genes etc. We have modified the related sentence in discussion section. Please see line 531-534 in the revised ms.

Reference:

Suzuki, Y., Baidaliuk, A., Miesen, P., Frangeul, L., Crist, A.B., Merklings, S.H., Fontaine, A., Lequime, S., Moltini-Conclois, I., Blanc, H., et al. (2020). Non-retroviral Endogenous Viral Element Limits Cognate Virus Replication in *Aedes aegypti* Ovaries. *Curr. Biol.* 30, 3495-3506 e3496. 10.1016/j.cub.2020.06.057.

4. The cellular function of NIETLVE14 was studied through knockout it by CRISPR/Cas9, the evidence will be more powerful if the prolonged development and decreasing hatching rate in mutation strain could be recovered by knock in NIETLVE14.

Response 4: Thank you for your valuable suggestions. Yes, we do agree that the evidence provided by knock in of NIETLVE14 will consolidate the current results. However, to the best of our knowledge, knock in technology has not been established in planthopper species and we are also trying to set up this powerful system. In our study, the cellular function of NIETLVE14 was investigated using both RNAi and knockout (CRISPR/Cas9) approaches, and each method indicated the consistent results with sufficient replicates, confirming that the observed phenotypes were reliable.

5. NIETLVE14 was relatively high abundance in the carcass and wing buds at both transcription and protein levels. Do the authors observe the wing development in the NIETLVE14 mutant strain? Is the development postponed related with the regulating of JH or MH?

Response 5: Thank you for your valuable suggestion. We have conduct further experiments to compare the wing development between the stains of WT and NIETLVE14 mutants (KO-M1 and KO-M2). Our results showed that NIETLVE14 did not affect the insect wing formation, and no significant difference in the proportion of short-winged/long-winged morph was observed between WT and NIETLVE14 mutants (KO-M1 and KO-M2). These results were added in

Supplemental Figure S7C and **Supplemental Figure S8I**. Please also refer to the related text in line **374-375**, **839-840** of the revised ms.

To investigate the potential relationship for the postponed development of planthopper with regulation of JH or MH, differentially expressed genes (DEGs) were determined between NIETLVE14 knockout mutant (KO-M1) and the control (WT). The results showed that DEGs are clearly associated with planthopper development, especially for the cuticle-associated genes (Supplemental Figure S11). Specifically, we found the expression of a ecdysteroids biosynthesis gene CYP302A1 and a neuroendocrine convertase were significantly affected in the KO-M1 strain (Supplemental Figure S11C), suggesting that hormonal pathways might also be involved in the regulation of the planthopper phenotype in the mutant strain. Nevertheless, whether the phenotype of the postponed development is regulated by hormonal pathways in *N. lugens* still needs further investigation. Please see the newly provided Supplemental Figure S11 and lines **404-417**, **899-915** of the revised ms.

REVIEWER COMMENTS

Reviewer #1 (Remarks to the Author):

Authors revised the ms, also by including additional data. However, to me the ms is still very confused for the order in which the data are presented, and the information provided. There are a lot of data, but often why such data was generated, and the experimental details do not make much sense. For instance, new transcriptomic data comparing the KO-MUT and the WT are presented, but the RNAseq analyses was done on nymphs...why? Importantly, from figure 3 (panel b), it looks like NIToEVE14 is transcribed as a longer mRNA than what predicted from the viral integration. Please specify where the 4 bp deletion occurs along the sequence of the NIToEVE14 in the KO-M1 strain. Is it inside the viral ORF within exon 3 or in exon 1 or exon 2? Can we consider this as a real domestication of a viral sequence? Or the viral sequence disrupted a planthopper gene and possibly altered its function? Because NIToEVE14 derives from the CP gene, please discuss the other EVE identified corresponding to this viral gene: how many have intact ORF? For how many did you have evidence of translation?

There are also still a lot of grammar errors that makes reading this article quite difficult. The article needs to be proofread by a native English speaker. Additionally, throughout the text, please avoid using "some ToEVes" and "certain ToEVes", instead put the precise number or list which ToEVes you are referring to. Also please avoid using expressions as "hasn't".

Additional comments:

Line 27: In the sentence "Some ToEVes display exon-intron structures and active", instead of some put the precise number.

Line 28 Please change the sentence "suggesting that they might evolve as intrinsic parts of planthopper mRNAs" to "suggesting that they might have been domesticated by planthoppers "

Line 33 the sentence "they may play significant roles in the functional diversity of arthropod genes" does not really make sense. Possibly: "they may contribute to the functional diversity of arthropod genes"?

Line 53. Most nrEVes have been identified from insects, please cite the most recent review summarising data on nrEVes in insects: Palatini et al, 2022 Current Opinion in Insect Sciences 49: 22-30.

Lines 116-118. Please justify why it would be valuable to look at the presence of integrations from totiviruses in three planthoppers? Why three, and which are these three species?

Lines 184-187. Sentence "It is also....in the opposite direction", please re-phrase as it is unclear how two ToEVes on different chromosomes can be located in the same chromosome of the planthopper genome.

Lines 204-208. This paragraph is very important as it correlates ToEVes phylogeny with the phylogeny of the three planthoppers species in which they were found. For this reason, please re-write the paragraph as the way it is written now is unclear.

Line 217. To have an overview OF

Lines 217-224. Please first describe the insect populations in which you analysed the ToEVes from and then the results of your analyses.

Line 236-237. I do not understand why the identification of a piggyBac element between NIToEVE3 and NIToEVE4 could suggest that these two EVes derived from the same viral insertion.

Line 240-241. Which evolutionary scenarios?

Line 242. Please elaborate on the finding of NiToEVE14 in *N. muii*. Were all the ToEVes identified in the three planthoppers species *N. lugens*, *L. striatellus*, *S. furcifera* searched for also in *N. muii*? What is the phylogenetic relationship between *N. muii* and *N. lugens*?

Lines 249-256. Please cite Pischedda et al., 2019. Front. Genet. 10:93.

Lines 282-285. Please check English.

Line 299 and lines 354-357. Stating "a peak of approximately 24-28 h" means a start is known. 24-48 h after what?

Lines 305-308. Please specify the origin of the transcriptomes. In previous paragraphs, you showed that not all ToEVes are found in all planthoppers populations, so the absence of a ToEVes may not mean absence of transcription.

Line 310. The word "contained" does not make sense here. What do you want to say?

Line 314-316. Please re-phrase. What does it mean that the location of ToEVs was within the predicted ORF or included the ORF?

Line 376-377. What do you call "offspring"? Please cite first eggs, then hatching rate. It is unclear if offspring refers to eggs or nymphs.

Lines 448-455. This is a significant result, but I would focus the analyses to orthologues of NiToEVE14 if any or ToEVs encompassing a viral ORF with the same protein motives as those identified in the NiToEVE14 ORF.

Line 458. NrEVs are not widely distributed, it really depends on the host genome and the viral species. Considering this ms is under consideration in a highly impactful and widespread journal, authors should strive to be as precise as possible. Additionally, the sentence "However, the potential functions and biological roles of evolutionally co-opted NIRVSNrEVs, especially at the protein level, remain largely unknown, despite the growing number of newly identified NIRVSNrEVs in arthropod genomes (Gilbert and Belliardo, 2022; Houe et al., 2019)" is not precise as most nrEVs in arthropod genomes do not encompass a complete ORFs so they could not be domesticated. The points here are that nrEVs are being identified across different eukaryotic genomes, their abundance and widespread is not homogenous, as it is highly dependent on the viral species and the host genomes. Also, most nrEVs are fragmented sequences and do not encompass a complete ORF in all genomes studied so far. A very limited number of nrEVs have been identified to encompass a complete ORF and be transcribed. This manuscript is the first to show that one nrEV is not only transcribed but also translated. This is the novelty. Please re-phrase the start of your discussion to clearly put results in the right perspective.

Line 468-470. Please put references to support this statement.

Overall, the discussion reads a summary of the results and not really as a discussion of the significance of the results in the context of current literature.

Lines 518-521. Please re-phrase.

Line 581. Palatini et al., 2022 did not state that arthropods with EVs are those closely related to the host of EVE-homology exogenous viruses. What Palatini said is that some arthropods harbour more EVs than others, independently from being host for viruses. For instance, Insect Specific Viruses readily infect Anopheles species, which do not harbour many nrEVs and also Aedes species, which instead have hundreds of nrEVs from different ISVs. Please correct.

Line 659. The title of this paragraph is not in line with the content of the paragraph, which talks about comparing the protein datasets across the three planthopper species studied here.

Lines 668-678. To which genome were the reads from *N. muii* aligned to? Why *N. muii* was not included in the overall analyses?

Lines 683-687. The use of different stringency criteria can bias results.

Lines 739-751. How do you distinguish transcription of ToEVs from infecting totivirus?

Lines 752-753. Were the infects used for RNA extraction tested for the absence of totivirus infection?

Lines 837 and 840. 50-60 replicates of how many insects each?

Lines 899-915. What is the significance of performing transcriptome analysis on the juvenile nymph stage when most differences between the WT and the KO-M1 strains were seen at the adult stage both in terms of longevity and fecundity?

Lines 1236-1237. Please use sense and antisense instead of same direction or opposite direction.

Line 1290. On which basis do you say that endogenization was a rare event

Reviewer #2 (Remarks to the Author):

The authors made revisions according to my suggestions, this MS could be accepted.

Reviewer #3 (Remarks to the Author):

The authors have added the supplementary experiments and revised the manuscript carefully according to reviewer's recommendation, they also explained the questions one by one in detail. I have no more suggestion on this manuscript and recommend to publish it in Nature Communication.

Reviewer #1 (Remarks to the Author):

Authors revised the ms, also by including additional data. However, to me the ms is still very confused for the order in which the data are presented, and the information provided. There are a lot of data, but often why such data was generated, and the experimental details do not make much sense. For instance, new transcriptomic data comparing the KO-MUT and the WT are presented, but the RNAseq analyses was done on nymphs...why?

Response: Thank you very much for your comments. We have carefully revised the ms following your valuable suggestions and comments. Please see the detailed response listed below. Regarding to the planthopper 5th instar nymphs used for comparative RNAseq analysis between KO-MUT and the WT, the reasons are as following:

- 1) A periodic expression of NIToEVE14 was observed during nymph stages, and the overall expression of NIToEVE14 was higher in nymphs than that of the adults (Fig.S4a, Fig.S5d);**
- 2) Development of planthopper nymph stages were significantly prolonged in NIToEVE14 KO-MUT (Fig.4b, Fig.S9b) or NIToEVE14-silenced (Fig.S10a) planthoppers compare to the WT or dsGFP-treated controls;**
- 3) NIToEVE14 potentially interacts with a planthopper molting-associated gene NITwd (Fig.S11a,b), which is not expressed in adult stage (Fig.S11d);**
- 4) The formation of planthopper ovaries began at the late stage of 5th instar nymph, therefore affects the number of eggs;**
- 5) The 5th instar nymph stage might also associate with the longevity difference between the KO-mutant and the WT.**

In the revised manuscript, we added the reasons for choosing 5th instar nymph as follow (Lines 935-940): “Considering the periodic expression of NIToEVE14 during nymph stages (Supplemental Figure S4a and S5d), a prolonged nymphal development after NIToEVE14 knockout/silencing (Fig. 4b; Supplemental Figure S9b; Supplemental Figure S10a), and a potential role of NIToEVE14 in insect molting (Supplemental Figure S11). The 5th instar nymphs of *N. lugens*, 72 h after molting, were selected for this analysis”.

Importantly, from figure 3 (panel b), it looks like NIToEVE14 is transcribed as a longer mRNA than what predicted from the viral integration. Please specify where the 4 bp deletion occurs along the sequence of the NIToEVE14 in the KO-M1 strain. Is it inside the viral ORF within

exon 3 or in exon 1 or exon 2? Can we consider this as a real domestication of a viral sequence? Or the viral sequence disrupted a planthopper gene and possibly altered its function?

Response: Thank you for this important comment and we do agree that this point is essential for the present work. We believe that the NIToEVE14 phenotype was caused by a genuine case of viral sequence domestication rather than by the disruption or alteration of a planthopper gene function. Our confidence in this conclusion is based on the following reasons:

1) Considering NIToEVE14 is located in the C-terminal of predicted ORF within the exon 3, the sgRNA in exon3 was designed to target the boundary region of *N. lugens*-derived sequence and viral integration sequence (Fig.3bii). As the result, two mutant strains (KO-M1 and KO-M2) were successfully obtained which retain the majority of planthopper-derived peptides of the regions 1-216aa and 1-218aa, respectively. Please see the newly provided Fig.S6a and the figure below.

2) It should be noted that although BlastP search result indicated the CP of *Sogatella furcifera* toti-like virus 2 (accession: WGU15424.1) has lowest e-value ($1e-70$) among all of the totiviruses, we found another totivirus, *Sanya nephotettix cincticeps* totivirus 1 (accession: UHK03213.1), exhibited homologous to NIETLVE14-m from 229aa to 926aa as indicated in the figure below. Therefore, the two mutant strains disrupt the entire eve region and our knockout experiments suggested that the phenotypes of the mutant planthoppers were caused by the domesticated viral sequence.

3) Also, protein-protein interaction assays demonstrated that the C-terminal of NIToEVE14 (NIToEVE14-C, 509-948aa), which located within the eve region (newly provided Fig.S6b), is responsible for the integration with NITwd, a molting related gene of planthopper.

Based on the above reasons, NIToEVE14 has played a role in insect physiology. In the revised manuscript, a schematic diagram was added (Fig.S6) showing the viral integration region, the mutation sites, and the regions selected for protein-protein

interaction assay. Accordingly, the relevant contents were added in result section (Lines 359-366), discussion section (Lines 533-545) and Figure legend (Lines 1344-1350).

Because NIToEVE14 derives from the CP gene, please discuss the other EVE identified corresponding to this viral gene: how many have intact ORF? For how many did you have evidence of translation?

Response: Thank you for your suggestion. Besides NIToEVE14, there are 18 planthopper ToEVES contain sequences derive from viral CP, including NIToEVE1, NIToEVE3, NIToEVE6, NIToEVE7-10, NIToEVE13, NIToEVE17-18, NIToEVE22, LsToEVE1, LsToEVE3, NIToEVE6-9, and SfToEVE1 (Fig.1a-c). Among these ToEVES, three ToEVES (NIToEVE13, LsToEVE1, and LsToEVE6) potentially contain intact ORFs (Fig.4b-c). Although transcripts of NIToEVE13, LsToEVE1, and LsToEVE6 were found in the transcriptomes (Fig.4a), we do not have direct evidence whether these three ToEVES can be translated into proteins. In the revised manuscript, the following related content has been added in the discussion (Lines 514 - 520): “It is worth noting that an additional three ToEVES (NIToEVE13, LsToEVE1, and LsToEVE6) derived from CP and one derived from RdRP (NIToEVE5) of totivirus potentially contain ORFs. While NIToEVE14 was the only peptide identified in the screened proteomic datasets, it’s important to note that the potential for translation of these ToEVES cannot be excluded, warranting further investigation.”

There are also still a lot of grammar errors that makes reading this article quite difficult. The article needs to be profred by a native English speaker. Additionally, throughout the text, please avoid using “some ToEVES” and “certain ToEVES”, instead put the precise number or list which ToEVES you are referring to. Also please avoid using expressions as “hasn’t”.

Response: Thank you for your suggestion and we apologize for any confusion resulting from language issues. We have taken your suggestion into account and have substantially improved the manuscript with Springer Nature Author Service (a professional language service recommended by Nature Communication) to polish English language and writing. All inaccuracies in the descriptions (including your suggestions) have been rectified throughout the manuscript. We believe that the revised English text now complies with the publication requirements of this journal.

Additional comments:

1. Line 27: In the sentence “Some ToEVEs display exon-intron structures and active”, instead of some put the precise number.

Response 1: Done (Line 26). “Some” was replaced with “Three”.

2. Line 28 Please change the sentence “suggesting that they might evolve as intrinsic parts of planthopper mRNAs” to “suggesting that they might have been domesticated by planthoppers ”

Response 2: Done (Lines 27-28).

3. Line 33 the sentence “they may play significant roles in the functional diversity of arthropod genes” does not really make sense. Possibly: “they may contribute to the functional diversity of arthropod genes”?

Response 3: Thank you for your suggestion and the sentence has been changed accordingly (Line 32).

4. Line 53. Most nrEVEs have been identified from insects, please cite the most recent review summarising data on nrEVEs in insects: Palatini et al, 2022 Current Opinion in Insect Sciences 49: 22-30.

Response 4: Thank you for your suggestion and the reference has been added accordingly (Lines 50-52).

5. Lines 116-118. Please justify why it would be valuable to look at the presence of integrations from totiviruses in three planthoppers? Why three, and which are these three species?

Response 5: We apologize that the background of the rice planthopper species has not been clarified in the instruction section. Actually, these three planthoppers are considered to be the most important rice pests in Asia through direct feeding and transmission of various rice viruses.

In the revised manuscript, the following instruction of the three planthoppers has been added (Lines 114 - 124): “The brown planthopper (*Nilaparvata lugens* (Stål)), white-backed planthopper (*Sogatella furcifera* (Horváth)) and small brown planthopper (*Laodelphax striatellus* (Fallén)) are three of the most destructive insect pests in the rice field belonging to the family Delphacidae. Recently, the availability

of chromosome-level genomes of the three planthoppers (Ma et al., 2021) and the discovery of novel totiviruses (Zhang et al., 2018) provided an opportunity to investigate the viral integration in these agriculturally important pests.”

6. Lines 184-187. Sentence “It is also...in the opposite direction”, please re-phrase as it is unclear how two ToEVes on different chromosomes can be located in the same chromosome of the planthopper genome.

Response 6: We are sorry for the confusion. In the revised manuscript, the sentence has been modified for clarity (Lines 180-184): “It is also noteworthy that NIToEVE3, NIToEVE4, and NIToEVE5 were located on the sense strand in Chr2 of *N. lugens*, while NIToEVE2 was located on the antisense strand of Chr2 (Fig.1a).”

7. Lines 204-208. This paragraph is very important as it correlates ToEVes phylogeny with the phylogeny of the three planthoppers species in which they were found. For this reason, please re-write the paragraph as the way it is written now is unclear.

Response 7: We are sorry for the confusion. In the revised manuscript, the sentence has been modified for clarity (Lines 199-209): “Therefore, the wide ToEVes integration of orthologous group-1 in three planthopper genomes suggested that the ancient viral integration events might have occurred before 64.4 million years ago. In contrast, ToEVes integration of orthologous group-3 was only detected in *N. lugens*, implying another independent viral integration event potentially occurred after 64.4 million years ago.”

8. Line 217. To have an overview OF

Response 8: Done (Line 214).

9. Lines 217-224. Please first describe the insect populations in which you analysed the ToEVes from and then the results of your analyses.

Response 9: Thank you for your suggestion. The insect populations used for ToEVes analysis were collected from different countries of Asia and a site from north Australia as indicated in our previous study (Hu et al., 2019). Please see Lines 212-214 in the revised manuscript.

Hu, Q.L., Zhuo, J.C., Ye, Y.X., Li, D.T., Lou, Y.H., Zhang, X.Y., Chen, X., Wang, S.L., Wang, Z.C., Lu, J.B., Mazlan N., Nguyen H.C., OO S.S., Thet T., Sharma P.N., Jauharlina, J., Rahman S.M.M.,

Ansari N.A., Chen A.D., Zhu Z.R., Heong K.L., Chen J.A., Zhan S., and Zhang C.X. (2019). Whole genome sequencing of 358 brown planthoppers uncovers the landscape of their migration and dispersal worldwide. *bioRxiv*, 798876.

10. Line 236-237. I do not understand why the identification of a piggyBac element between NiToEVE3 and NiToEVE4 could suggest that these two EVEs derived from the same viral insertion.

Response 10: Thank you for pointing out this. Previous studies indicated that the herpesvirus can be fused with a piggyBac-like DNA transposon and form a novel mobile element in vertebrate (Inoue et al., 2017, 2018). Thus, the piggyBac element identified between NiToEVE3 (C-terminal of totivirus CP) and NiToEVE4 (N-terminal of totivirus RdRP) might also associated with early totivirus insertion events. Nevertheless, more evidences are needed to support this conclusion in the future. In the revised manuscript, we have added this information and toned down the conclusion. Please see Lines 230-237. The two references were also provided accordingly in Lines 1077-1082 of the revised manuscript.

Inoue, Y., Saga, T., Aikawa, T., Kumagai, M., Shimada, A., Kawaguchi, Y., Naruse, K., Morishita, S., Koga, A., and Takeda, H. (2017). Complete fusion of a transposon and herpesvirus created the Teratorn mobile element in medaka fish. *Nat. Commun.* 8, 551. [10.1038/s41467-017-00527-2](https://doi.org/10.1038/s41467-017-00527-2).

Inoue, Y., Kumagai, M., Zhang, X., Saga, T., Wang, D., Koga, A., and Takeda, H. (2018). Fusion of piggyBac-like transposons and herpesviruses occurs frequently in teleosts. *Zoological letters* 4, 6. [10.1186/s40851-018-0089-8](https://doi.org/10.1186/s40851-018-0089-8).

11. Line 240-241. Which evolutionary scenarios?

Response 11: We assume that ToEVEs in *L. striatellus* and *S. furcifera* underwent different natural selection during the long-term co-evolution within its insect hosts. So the evolutionary scenarios refer to host selection such as positive selection or negative selection. Please see Line 239 in the revised manuscript.

12. Line 242. Please elaborate on the finding of NiToEVE14 in *N. muiri*. Were all the ToEVEs identified in the three planthoppers species *N. lugens*, *L. striatellus*, *S. furcifera* searched for also in *N. muiri*? What is the phylogenetic relationship between *N. muiri* and *N. lugens*?

Response 12: Thank you for your suggestion. *N. lugens* and *N. muiri* belong to the same genus in the family Delphacidae. Currently, high-quality genome of *N. muiri* is

still not available and only genome resequencing reads from two individuals were available (Hu et al., 2019). Yes, all of the identified planthopper ToEVEs were searched against the two individual genomes of *N. muiri*, and the results showed that only NIToEVE13 and NIToEVE14 were detected with 98.7-100% coverage in *N. muiri*. Please see Lines 240-246 in the revised manuscript.

Hu, Q.L., Zhuo, J.C., Ye, Y.X., Li, D.T., Lou, Y.H., Zhang, X.Y., Chen, X., Wang, S.L., Wang, Z.C., Lu, J.B., Mazlan N., Nguyen H.C., OO S.S., Thet T., Sharma P.N., Jauharlina, J., Rahman S.M.M., Ansari N.A., Chen A.D., Zhu Z.R., Heong K.L., Chen J.A., Zhan S., and Zhang C.X. (2019). Whole genome sequencing of 358 brown planthoppers uncovers the landscape of their migration and dispersal worldwide. bioRxiv, 798876.

13. Lines 249-256. Please cite Pischedda et al., 2019. Front. Genet. 10:93.

Response 13: Done (Line 256).

14. Lines 282-285. Please check English.

Response 14: Sorry for the confusion. This sentence has been modified for clarity (Lines 280-286): “The discrepant expression patterns of the same ToEVEs in various planthopper datasets suggest that these ToEVEs were absent in some of the planthopper datasets, as illustrated in Fig.2a. Alternatively, another explanation could be that the corresponding transcripts of these ToEVEs are only induced under specific circumstances.”

15. Line 299 and lines 354-357. Stating “a peak of approximately 24-28 h” means a start is known. 24-48 h after what?

Response 15: Sorry for the confusion. Here “a peak of approximately 24-28 h” means “a peak of approximately 24-28 h after molting”. In the revised manuscript, the information has been indicated accordingly. Please see Lines 299, 349, and 352.

16. Lines 305-308. Please specify the origin of the transcriptomes. In previous paragraphs, you showed that not all ToEVEs are found in all planthoppers populations, so the absence of a ToEVEs may not mean absence of transcription.

Response 16: Thank you for your valuable suggestion. We do agree that the absence of a ToEVEs may not mean absence of transcription. This sentence has been modified accordingly (Lines 305-312): “To better understanding the transcription of ToEVEs, the potentially transcribed ToEVEs shown in Fig.3a (red fonts) were choosed. The corresponding transcriptomes with high abundant ToEVEs transcripts (Fig.3a and Supplemental Figure S4) were selected, assembled and further characterized (Fig.3b-d).” The origin of corresponding transcriptome was provided with accession number for each of the ToEVE (Fig.3b-d).

17. Line 310. The word “contained” does not make sense here. What do you want to say?

Response 17: Sorry for the confusion. The sentence has been modified for clarity (Lines 312-314): “Most of the assembled planthopper transcripts were longer than the ToEVEs (except LsToEVE9, Fig.3c-vi) and had relatively high coverage abundance (Fig.3b-d).”

18. Line 314-316. Please re-phrase. What does it mean that the location of ToEVEs was within the predicted ORF or included the ORF?

Response 18: Sorry for the confusion. The sentence has been modified for clarity (Lines 316-318): “Furthermore, ToEVEs were found within the ORFs (NiToEVE13, NiToEVE14, and LsToEVE1), encompassed the ORF (NiToEVE5), or were not associated with the ORF (LsToEVE6).”

19. Line 376-377. What do you call “offspring”? Please cite first eggs, then hatching rate. It is unclear if offspring refers to eggs or nymphs.

Response 19: Sorry for the confusion. Here “offspring” refers to “nymphs”. Therefore, the “offspring” has been replaced with “nymphs” in Fig.4 and Fig.S9. The orders of eggs, hatching rate, nymphs have been adjusted as suggested. Please see Lines 375-377, 1288-1290, 1368-1370 in the revised manuscript, as well as the revised Fig.4 and Fig.S9.

20. Lines 448-455. This is a significant result, but I would focus the analyses to orthologues of NiToEVE14 if any or ToEVEs encompassing a viral ORF with the same protein motives as those identified in the NiToEVE14 ORF.

Response 20: Thank you for your suggestion. We agree with the point and have already done this analysis. Our result showed that there is no NIToEVE14 orthologues presented in these ToEVEs. So these results were not provided in the current work.

21. Line 458. NrEVEs are not widely distributed, it really depends on the host genome and the viral species. Considering this ms is under consideration in a highly impactful and widespread journal, authors should strive to be as precise as possible. Additionally, the sentence “However, the potential functions and biological roles of evolutionally co-opted NIRVSnrEVEs, especially at the protein level, remain largely unknown, despite the growing number of newly identified NIRVSnrEVEs in arthropod genomes (Gilbert and Belliardo, 2022; Houe et al., 2019)” is not precise as most nrEVEs in arthropod genomes do not encompass a complete ORFs so they could not be domesticated. The points here are that nrEVEs are being identified across different eukaryotic genomes, their abundance and widespread is not homogenous, as it is highly dependent on the viral species and the host genomes. Also, most nrEVEs are fragmented sequences and do not encompass a complete ORF in all genomes studied so far. A very limited number of nrEVEs have been identified to encompass a complete ORF and be transcribed. This manuscript is the first to show that one nrEVE is not only transcribed but also translated. This is the novelty. Please re-phrase the start of your discussion to clearly put results in the right perspective.

Response 21: Thank you very much for your valuable comments and we are sorry for the inaccurate description. We have carefully rephrased the start of the discussion following your suggestion to avoid the confusion. In the revised manuscript, the description was modified as following (Lines 454-460): “NrEVEs, a recently discovered type of EVE, have been characterized in various eukaryotic genomes, especially in insect genomes such as mosquitoes (Horie et al., 2010; Wallau, 2022). The abundance and distribution of nrEVEs are not homogenous, as they depend significantly on the viral species and host genomes (Palatini et al., 2022). However, the potential functions and biological roles of evolutionarily co-opted nrEVEs, especially at the protein level, remain largely unknown at present (Gilbert and Belliardo, 2022; Houe et al., 2019).”

22. Line 468-470. Please put references to support this statement.

Response 22: Done (Line 468).

23. Overall, the discussion reads a summary of the results and not really as a discussion of the significance of the results in the context of current literature.

Response 23: Thank you for your comment, and we have made the necessary revisions to better highlight the significance of our work. The key discovery of this study is that we proved nrEVEs can be domesticated in arthropods and function at the protein level. Therefore, our discussions primarily focused on this finding, supported by the present results, including the distribution of ToEVEs in planthopper populations, the expression of ToEVEs in planthoppers, and the role of NIToEVE14 in planthopper biology. In comparison to previous studies, we emphasize the significance of our research and its potential contributions to future work in this field. Notably, our extensive screening work suggests that the extent of domesticated ToEVEs may be currently underestimated.

24. Lines 518-521. Please re-phrase.

Response 24: Done. This sentence has been rephrased as following (Lines 514-520): “It is worth noting that an additional three ToEVEs (NIToEVE13, LsToEVE1, and LsToEVE6) derived from CP and one derived from RdRP (NIToEVE5) of totivirus potentially contain ORFs. While NIToEVE14 was the only peptide identified in the screened proteomic datasets, it’s important to note that the potential for translation of these ToEVEs cannot be ruled out, warranting further investigation.”

25. Line 581. Palatini et al., 2022 did not state that arthropods with EVEs are those closely related to the host of EVE-homology exogenous viruses. What Palatini said is that some arthropods harbour more EVEs than others, independently from being host for viruses. For instance, Insect Specific Viruses readily infect Anopheles species, which do not harbour many nrEVEs and also Aedes species, which instead have hundreds of nrEVEs from different ISVs. Please correct.

Response 25: Thank you for your comment and we apologize for the incorrect reference. The reference was removed and the sentence was rephrased. Please see Lines 582-584 in the revised manuscript.

26. Line 659. The title of this paragraph is not in line with the content of the paragraph, which talks about comparing the protein datasets across the three planthopper species studied here.

Response 26: Thank you for your suggestion. The title has been changed to “Orthologous analysis for protein datasets of the three planthopper species” (Lines 659-660).

27. Lines 668-678. To which genome were the reads from *N. muiri* aligned to? Why *N. muiri* was not included in the overall analyses?

Response 27: Thank you for your comments. Currently, no high-quality genome is available for *N. muiri*. So ToEVes in the genome of *N. muiri* are not identified yet and this species was not included in the overall analysis. And the identified ToEVes derived from the other three planthoppers were used as query and searched against the genome resequencing reads of *N. muiri*. The related content has been modified as following: (Lines 671-676) “The genome resequencing reads of *N. lugens*, *L. striatellus*, and *S. furcifera* were then mapped to a collection of the identified planthopper ToEVes using BWA MEM version 0.7.17 (Li and Durbin, 2009). For *N. muiri*, considering that there is no high-quality genome currently available for this species, ToEVes identified in three planthoppers were searched against the genome resequencing reads of *N. muiri*.”

Please also refer to the **Response 12**.

28. Lines 683-687. The use of different stringency criteria can bias results.

Response 28: Thank you for pointing out this. Different stringency criteria used here is mainly because the genome sequencing depth for three planthopper species is different. As indicated in the manuscript, the average depth of *L. striatellus* and *S. furcifera* was 31.5× and 35.0×, respectively, while the average sequencing depth for *N. lugens* was only 11.5× (Lines 683-687). Moreover, the presence of ToEVes was compared among different insect population of the same species rather than different species. Therefore, the different stringency criteria between species might not affect the variable distribution of ToEVE in the populations of the same species.

29. Lines 739-751. How do you distinguish transcription of ToEVes from infecting totivirus?

Response 29: Thank you for your valuable comments. We do agree that it's essential to precisely distinguish ToEVes from infecting totiviruses. In our case, the identified ToEVes and planthopper-infecting totiviruses share no similarity at the nucleic acid level. And BlastN search against genomes of the three planthoppers gives not significant hits using the collection of infecting totivirus genomes.

30. Lines 752-753. Were the infects used for RNA extraction tested for the absence of totivirus infection?

Response 30: Yes, the insects used in this study were conformed to be absence of totivirus infection. The information has been indicated in the revised manuscript accordingly (Lines 751-752).

31. Lines 837 and 840. 50-60 replicates of how many insects each?

Response 31: We are sorry for the inaccurate description. This information has been modified as following (Lines 843-847): “For each strain, 50-60 individual insects were used for survival, developmental duration, and adult longevity analysis; 150-200 individual insects were used for female-male ratio analysis; 200-300 individual insects were used for short-winged/long-winged analysis; 15-20 individual insects were used for fecundity analysis.”

32. Lines 899-915. What is the significance of performing transcriptome analysis on the juvenile nymph stage when most differences between the WT and the KO-M1 strains were seen at the adult stage both in terms of longevity and fecundity?

Response 32: Please refer to our first response providing five reasons why 5th instar nymphs of planthopper were chosen for this analysis. And relevant content was added in the revised manuscript (Lines 901-906).

33. Lines 1236-1237. Please use sense and antisense instead of same direction or opposite direction.

Response 33: Done (Lines 1246-1248).

34. Line 1290. On which basis do you say that endogenization was a rare event

Response 34: Sorry for the inaccurate description. We have removed “by rare events” (Line 1301).

REVIEWERS' COMMENTS

Reviewer #1 (Remarks to the Author):

I appreciate authors extensively revising the ms. Now the ms reads well and, most importantly, is clear and focused.

I just few additional small comments.

1) Line 283-Line287: Please specify whether you are referring to N. lugens or viral ORFs. To me the revised sentence "Furthermore, ToEVes were found within the ORFs (NIToEVE13, NIToEVE14, and LsToEVE1), encompassed the ORF (NIToEVE5), or were outside the ORF (LsToEVE6)" is misleading. Your focus has been ToEVes through the text. What you have to show is whether the viral ORF bioinformatically-predicted for your ToEVE is actually expressed or not. What you show in Fig. 3b is that NToEVE13 and NToEVE14 are annotated within exon 2 or exon 3 of N. lugens genes and are expressed as part of these genes. The same for lines 287-289: splicing regards N. lugens genes, it has nothing to do with the exon containing the viral integration. So in the discussion (line 463 and line 473) the expression "intron containing EVE" is not precise as ToEVes do not have any intron. Unless you find a N. lugens without NToEVE14, with expression in the gene where integration occurred, you cannot say that exon1-2 were "recruited" after integration to be part of the coding sequence that includes ToEVes or not.

2) Line 477: please change "it's" to "it is"

3) If allowed, I would suggest having fig. S5 as part of the main manuscript as the demonstration that NIToEVE14 is part of a translated protein is really the novelty of this ms.

4) I am not convinced that you demonstrate that NIToEVE14 provides a novel cellular function. What you show is that this viral integration has been domesticated to provide a biological function, that this function is novel as you claim in the title is a bit of stretch. So I would suggest changing the title to "Co-option of a non-retroviral endogenous viral element in planthoppers"

Reviewer #1 (Remarks to the Author):

I appreciate authors extensively revising the ms. Now the ms reads well and, most importantly, is clear and focused.

Response: Thank you very much for your critical and professional comments on this manuscript.

I just few additional small comments.

1) Line 283-Line287: Please specify whether you are referring to N. lugens or viral ORFs. To me the revised sentence "Furthermore, ToEVes were found within the ORFs (NIToEVE13, NIToEVE14, and LsToEVE1), encompassed the ORF (NIToEVE5), or were outside the ORF (LsToEVE6)" is misleading. Your focus has been ToEVes through the text. What you have to show is whether the viral ORF bioinformatically-predicted for your ToEVE is actually expressed or not. What you show in Fig. 3b is that NToEVE13 and NToEVE14 are annotated within exon 2 or exon 3 of N. lugens genes and are expressed as part of these genes. The same for lines 287-289: splicing regards N. lugens genes, it has nothing to do with the exon containing the viral integration. So in the discussion (line 463 and line 473) the expression "intron containing EVE" is not precise as ToEVes do not have any intron. Unless you find a N. lugens without NToEVE14, with expression in the gene where integration occurred, you cannot say that exon1-2 were "recruited" after integration to be part of the coding sequence that includes ToEVes or not.

Response: Thank you for this important comment and we are sorry for the imprecise description. In the revised manuscript, we have carefully distinguished ORFs originated from the insect and virus. Lines 283-289 have been revised as: "ORF prediction indicated that three NIToEVes (NIToEVE5, NIToEVE13, and NIToEVE14) and two LsToEVes (LsToEVE1 and LsToEVE6) were located within planthopper transcripts with intact ORFs ranging from 528 nt to 4737 nt (Fig.3b-c). Furthermore, NIToEVE13, NIToEVE14 and LsToEVE1 were annotated within exons of planthopper genes and are expressed as part of these genes." The discussion part has also been revised accordingly. Please see Line 270-274 and 439-440 in the revised manuscript.

2) Line 477: please change "it's" to "it is"

Response: Done (Line 453 in the revised manuscript).

3) If allowed, I would suggest having fig. S5 as part of the main manuscript as the demonstration that NIToEVE14 is part of a translated protein is really the novelty of this ms.

Response: Done. We moved Fig. S5 to main Fig. 4 of the revised MS.

4) I am not convinced that you demonstrate that NIToEVE14 provides a novel cellular function. What you show is that this viral integration has been domesticated to provide a biological function, that this function is novel as you claim in the title is a bit of stretch. So I would suggest changing the title to "Co-option of a non-retroviral endogenous viral element in planthoppers"

Response: Done. The manuscript title has been changed as you suggested. Thank you for all your helps.